# Tracing the Representation Geometry of Language Models from Pretraining to Post-training

**Melody Zixuan Li**[* 1 2]     **Kumar Krishna Agrawal**[* 3]     **Arna Ghosh**[* 1 2 10]

**Komal Kumar Teru** [4]     **Adam Santoro**[† 5]     **Guillaume Lajoie** [2 6 10]

**Blake A. Richards** [1 2 7 8 9 10]

## Abstract

Standard training metrics like loss fail to explain the emergence of complex capabilities in large language models. We take a spectral approach to investigate the geometry of learned representations across pretraining and post-training, measuring effective rank (RankMe) and eigenspectrum decay ($\alpha$ReQ). With OLMo (1B-7B) and Pythia (160M-12B) models, we uncover a consistent non-monotonic sequence of three geometric phases during autoregressive pretraining. The initial "warmup" phase exhibits rapid representational collapse. This is followed by an "entropy-seeking" phase, where the manifold's dimensionality expands substantially, coinciding with peak n-gram memorization. Subsequently, a "compression-seeking" phase imposes anisotropic consolidation, selectively preserving variance along dominant eigendirections while contracting others, a transition marked with significant improvement in downstream task performance. We show these phases can emerge from a fundamental interplay of cross-entropy optimization under skewed token frequencies and representational bottlenecks ($d \ll |\mathcal{V}|$). Post-training further transforms geometry: SFT and DPO drive "entropy-seeking" dynamics to integrate specific instructional or preferential data, improving in-distribution performance while degrading out-of-distribution robustness. Conversely, RLVR induces "compression-seeking", enhancing reward alignment but reducing generation diversity.

## 1   Introduction

Loss curves during training offer an incomplete account of how large language models (LLMs) learn specific behaviors [Wei et al., 2022, Ganguli et al., 2022]. While training loss typically decreases monotonically [Kaplan et al., 2020, Hoffmann et al., 2022], model capabilities and internal representational structures exhibit significant qualitative shifts [Singh et al., 2023, Brown et al., 2023, Singh et al., 2024]. This disconnect highlights a fundamental challenge: How do high-dimensional distributed representations within LLMs evolve during training, and how do these representational transformations give rise to emergent capabilities?

We answer this question by using spectral analysis to quantify the geometric evolution of LLM representations. We discover that this evolution is not a smooth progression but a consistent, three-

---

[*]Equal Contribution, [†] Advisory capacity only,  Correspondence: blake.richards@mcgill.ca
[1]Computer Science, McGill University, Canada  [2]Mila - Quebec AI Institute, Canada  [3]UC Berkeley, USA
[4]Cohere, Canada  [5]Google Deepmind, Canada  [6]Mathematics and Statistics, Université de Montréal, Canada
[7]Neurology & Neurosurgery, McGill University, Canada     [8]Montreal Neurological Institute, Canada
[9]CIFAR Learning in Machines & Brains Program, Canada  [10]Google Paradigms of Intelligence, Canada

39th Conference on Neural Information Processing Systems (NeurIPS 2025).

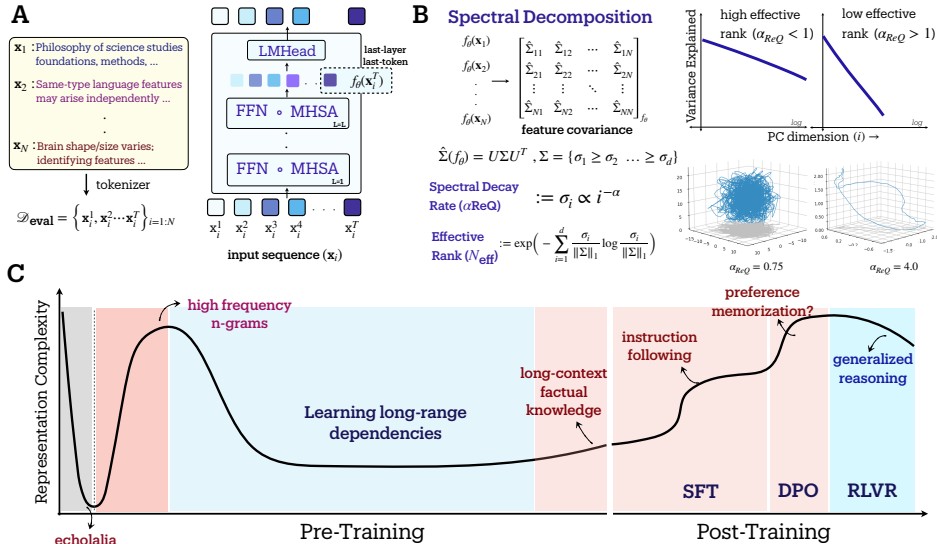

Figure 1: **Spectral framework reveals three universal phases in LLM training.** (A) LLM representations analyzed via empirical feature covariance $\hat{\Sigma}(f_\theta)$ of last-token hidden states $f_\theta(x_i)$. (B) Two complementary spectral metrics: $\alpha$ReQ measures eigenspectrum decay rate (variance concentration), while RankMe quantifies effective rank (utilized dimensionality). (C) Pretraining exhibits three phases: "warmup" (rapid collapse), "entropy-seeking" (2-3× expansion coinciding with n-gram memorization), and "compression-seeking" (anisotropic consolidation enabling long-context understanding). Post-training continues these dynamics: SFT/DPO induce "entropy-seeking" while RLVR induces "compression-seeking".

phase dynamic. Our method centers on the spectral properties of the covariance matrix of last-token representations, which capture rich information about the model's internal representations, especially when using causal attention. To measure this geometric structure, we compute two metrics from the eigenspectrum of these matrices: the effective rank ("RankMe"), and the power-law decay rate ("$\alpha$ReQ") of the eigenvalues [Garrido et al., 2023, Agrawal et al., 2022]. These spectral measures of representation geometry have been linked theoretically and experimentally to generalization in downstream tasks [Bartlett et al., 2020, Thilak et al., 2023]. Intuitively, representation geometry tells us about the model's expressive capacity, utilization, and amount of data compression.

Our analysis shows that LLM pretraining unfolds through a consistent sequence of distinct geometric phases marked by non-monotonic evolution of spectral properties. These phases correlate with significant shifts in the model's expressive power and information compression ability (c.f. Figure 1):

- An initial **"warmup"** phase, coinciding with learning rate ramp-up, where there is a rapid collapse of representations onto dominant data manifold directions.
- An **"entropy-seeking"** phase marked by manifold expansion in many directions, which correlates with an increase in n-gram distributional memorization.
- A **"compression-seeking"** phase with anisotropic consolidation along principal feature eigenvectors shows enhanced learning of long-range dependencies and robust generalization.

We further develop mechanistic insights from analytically tractable toy models, demonstrating that these geometric phase transitions are influenced by the interplay of cross-entropy optimization, information bottlenecks, and skewed data distributions.

Our investigation of post-training stages reveals analogous geometric shifts: Supervised fine-tuning (SFT) produces an "entropy-seeking"-like expansion with concomitant assimilation of specific instructions. Reinforcement Learning from Verifiable Rewards (RLVR) produces a "compression-seeking"-like contraction, which can consolidate reward-aligned behaviors at the cost of curtailing generative novelty and exploration. Together, our findings present a more granular view of LLM training, and offer practical implications for optimizing LLM training and adaptation pipelines based on desired downstream outcomes.

## 2 Methods

### 2.1 Spectral Analysis, Matrix Entropy, and Effective Rank

**Last token representations in autoregressive language models**: A rigorous understanding of LLM capabilities necessitates a precise characterization of the *geometry of their learned representations*. An autoregressive language model processes an input sequence of discrete tokens $\mathbf{s} = (t_1, t_2, \ldots, t_N)$, transforming each token $t_k$ through its $l$ layers (conditioned on preceding tokens $t_{<k}$) into a sequence of high-dimensional continuous vectors $\mathbf{f}_\theta^{(l)}(t_k|t_{<k})$. For autoregressive models, the representation of the final token ($t_N$) at the last layer, $\mathbf{y}_N := \mathbf{f}_\theta^{(L)}(t_N|t_{<N})$, is particularly pivotal. Its significance stems from different factors: (i) it directly parameterizes the predictive distribution for the subsequent tokens $P(t_{N+1}|t_1, ..., t_N)$; (ii) it synthesizes information from the entire context $t_{\leq N}$ to inform this prediction, meaning it inherently reflects the model's capacity for contextual understanding; and (iii) is often used as input to task-specific layers in downstream applications.

**High-dimensional representation complexity metrics**: To quantitatively measure representation geometry, we perform spectral analysis of the feature covariance matrix. Given a set of $M$ input sequences, we form a feature matrix $\mathbf{F} \in \mathbb{R}^{M \times d}$; each row is a feature vector of the last token $\mathbf{y}_N$ for each input. Assuming the features are centered, the empirical covariance matrix is $\hat{\Sigma} := \frac{1}{M}\mathbf{F}^T\mathbf{F}$. The eigenspectrum of $\hat{\Sigma}$, denoted by eigenvalues $\{\sigma_i(\hat{\Sigma})\}_{i=1}^d$, measures the concentration of information along the principal axes of variation. The distribution of $\{\sigma_i\}_{i=1}^d$ provides a quantitative description of feature geometry: a sharp decay indicates information compressed in a lower-dimensional subspace (anisotropic geometry), while a slow decay indicates a high-dimensional subspace is utilized.

This spectral perspective motivates using *matrix entropy* to measure the uniformity of the eigenvalue distribution. If $p_i = \sigma_i/(\sum_j \sigma_j)$ is the proportion of variance along the $i$-th principal axis, the Von Neumann *entropy-based effective rank* [Roy and Vetterli, 2007, Garrido et al., 2023] is defined as:

$$\texttt{RankMe} := \exp\left(S(\hat{\Sigma})\right) = \exp\left(-\sum_{i=1}^d p_i \ln p_i\right) \quad \in (0, d]. \tag{1}$$

Low entropy indicates a skewed eigenvalue distribution, i.e. low-dimensional (anisotropic) representations, while high entropy implies a uniform spread, i.e. high-dimensional (isotropic) representations.

Our empirical studies also show that LLM activation matrices exhibit *heavy-tailed* eigenvalue spectra, i.e., a power law distribution where $\sigma_i \propto i^{-\alpha\text{ReQ}}$, where $\alpha\text{ReQ} > 0$ [Ghosh et al., 2022]. Slower decay or smaller $\alpha\text{ReQ}$ implies a more uniform spread of $\sigma_i$'s (higher dimensional), and thus higher $S(\hat{\Sigma})$ and $\texttt{RankMe}$. Conversely, faster decay or larger $\alpha\text{ReQ}$ implies representations are compactly packed along fewer principal directions [Stringer et al., 2019, Agrawal et al., 2022], yielding lower entropy and smaller $\texttt{RankMe}$. $\alpha\text{ReQ}$ and $\texttt{RankMe}$ thus provide related metrics of representation geometry, though unlike $\texttt{RankMe}$, $\alpha\text{ReQ}$ does not change with the model's feature dimensionality, $d$.

### 2.2 Quantifying Distributional Memorization and Generalization via n-gram Alignment

To dissect how LLMs utilize their pretraining corpus $\mathcal{D}$, we differentiate *distributional memorization*, i.e. how aligned are LLM output probabilities with n-gram frequencies in $\mathcal{D}$, from *distributional generalization*, i.e. LLM capabilities beyond such statistics [Liu et al., 2024]. To quantify the alignment with n-gram statistics, we use the $\infty$-gram language model (LM) which uses the largest possible value of $n$ for predicting the next token probability. Briefly, an $\infty$-gram LM can be viewed as a generalized version of an $n$-gram LM which starts with $n = \infty$, and then performs backoff till the $n$-gram count in $\mathcal{D}$ is non-zero [Liu et al., 2024]. Consequently, the output probability of the $\infty$-gram LM for each token is dependent on its longest existing prefix in $\mathcal{D}$.

The distributional memorization metric is defined as the spearman rank correlation ($\rho_s$) between the $\infty$-gram LM outputs and the LLM outputs for all tokens in a target sequence [Wang et al., 2025]. Formally, consider a concatenated sequence of instructions, $u$, question, $x$ and target, $y$, from a question-answering task, $\mathcal{T}$. Then, the distributional memorization is computed as:

$$Mem_\infty(LLM, \mathcal{D}, \mathcal{T}) := \rho_s\left(\bar{P}_{\infty,\mathcal{D}}(y|u \oplus x), \bar{P}_{LLM}(y|u \oplus x)\right) \tag{2}$$

where $\bar{P}_\cdot(y|u \oplus x) := \prod_{t_i \in y} P_\cdot(t_i|u \oplus x \oplus y_{[t_0:t_{i-1}]})$ denotes the joint likelihood of all tokens in $y$ and $P_\cdot()$ is the next token prediction distribution, as described above.

## 2.3 Post-Training Methodologies and Evaluation

**Supervised Fine-Tuning (SFT)** adapts pre-trained LLMs by further training on a curated dataset $\mathcal{D}_{\text{SFT}} = \{(x_i, y_i)\}_{i=1}^{N_{\text{SFT}}}$ typically consisting of instruction-response pairs. The standard objective is to minimize the negative log-likelihood of the target responses, effectively maximizing $P_\theta(y|x)$ for examples in $\mathcal{D}_{\text{SFT}}$. We evaluate the robustness of the SFT model by contrasting its performance on held-out examples from $\mathcal{D}_{\text{SFT}}$ (In-Distribution, ID) with its performance on examples from a related but distinct dataset $\mathcal{D}_{\text{OOD}}$ (Out-of-Distribution, OOD), which may vary in task, style, or complexity not present in $\mathcal{D}_{\text{SFT}}$ [Springer et al., 2025].

**Preference Alignment and Reasoning** : For alignment beyond SFT, we consider Direct Preference Optimization (DPO) [Rafailov et al., 2023] and Reinforcement Learning from Verifiable Rewards (RLVR). DPO refines an LLM policy $\pi_\theta$ based on a static dataset of human preferences $\mathcal{D}_{\text{pref}} = \{(x, y_w, y_l)\}$, where the response $y_w$ is preferred over $y_l$ for prompt $x$. It directly optimizes for preference satisfaction by minimizing the loss:

$$\mathcal{L}_{\text{DPO}}(\pi_\theta; \pi_{\text{ref}}) = -\mathbb{E}_{(x,y_w,y_l)\sim\mathcal{D}_{\text{pref}}}\left[\log \sigma\left(\hat{r}_\theta(x, y_w) - \hat{r}_\theta(x, y_l)\right)\right], \tag{3}$$

where $\hat{r}_\theta(x, y) = \beta \log(\pi_\theta(y|x)/\pi_{\text{ref}}(y|x))$ represents the implicit log-ratio of probabilities scaled by $\beta$ against a reference policy $\pi_{\text{ref}}$, and $\sigma(.)$ is the logistic function. Reinforcement Learning from Verifiable Rewards (RLVR), as applied in works like Lambert et al. [2024] and Shao et al. [2024], optimizes the LLM's policy $\pi_\theta$ to maximize the expected discounted cumulative reward, $J(\theta) = \mathbb{E}_{\tau\sim\pi_\theta}\left[\sum_{t=0}^{T} \gamma^t R_t\right]$, where $\tau = (s_0, a_0, \ldots, s_T, a_T)$ is a trajectory generated by actions $a_t \sim \pi_\theta(\cdot|s_t)$ in states $s_t$, $\gamma \in [0, 1]$ is a discount factor, and $R_t = R(s_t, a_t)$ is the reward at time $t$. This optimization is typically performed using policy gradient algorithms (e.g., PPO). Critically, the reward $R_t$ in RLVR is derived from verifiable properties of the LLM's outputs, e.g. correctness on mathematical problems or passing unit tests.

**Performance with pass@k**: To evaluate problem-solving efficacy and generative exploration, particularly for RLVR-tuned models, we employ the pass@k metric [Kulal et al., 2019]. For a given problem, k independent responses are stochastically generated from the model; the problem is deemed solved if at least one response constitutes a verifiable solution. Since direct estimation of pass@k can exhibit high variance, we utilize the unbiased estimator Chen et al. [2021], Yue et al. [2025]:

$$\texttt{pass@k} = \mathbb{E}_{P_i}\left[1 - \frac{\binom{N-c_i}{k}}{\binom{N}{k}}\right] \tag{4}$$

where, N samples are generated for each problem $P_i$ , and $c_i$ denotes the count of correct solutions among them (parameters for this work are N=512 and k $\leq$ 256).

## 3 Probing the representation geometry of language models

To study LLM representation geometry at intermediate stages of the training lifecycle, we analyze checkpoints from three publicly released model suites. For all experiments, we used 15000 sequences from the the FineWeb sample-10BT dataset to probe the geometry of model representations. We defer additional details on the model architecture, dataset and training run to Appendix A.

- **OLMo framework** Groeneveld et al. [2024], OLMo et al. [2024], Lambert et al. [2024]: Developed by AI2, OLMo & OLMo-2 family of models provide intermediate checkpoints across different model sizes – 1B, 7B and 13B. We focused on intermediate checkpoints available for the OLMo-2 7B and 1B models throughout their $\sim 4T$ token training run.

- **Pythia suite** Biderman et al. [2023]: Developed by EleutherAI, this suite consists of models ranging from 70M to 12B parameters, all trained on the Pile dataset [Gao et al., 2020] using the same data ordering and hyperparameters across scales. We analyzed the intermediate checkpoints available at various intermediate training steps for 1B+ models.

- **Tülu-3.1 models** [Wang et al., 2024]: Developed by AI2, this suite contains instruction following $8B$ LLaMA-based models parameters, that were post-trained with state-of-the-art recipes. We analyzed checkpoints from all post-training stages of the model.

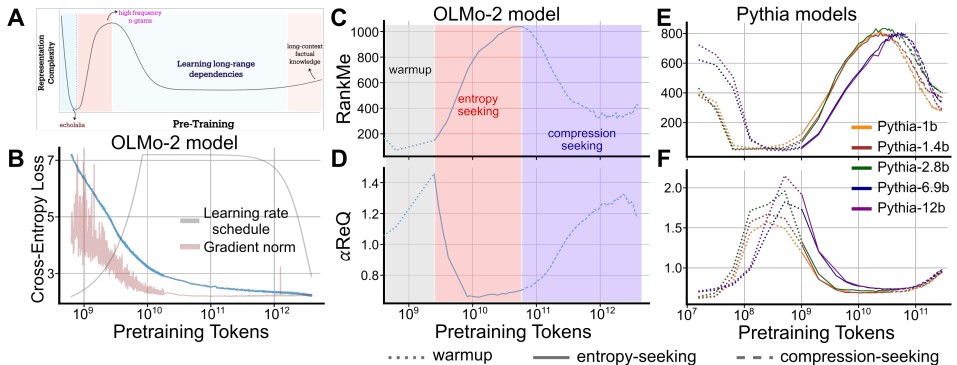

Figure 2: **Loss decreases monotonically, but representation geometry does not. (A)** Schematic from Fig 1, for the pretraining stage. **(B)** Cross-entropy loss, gradient norm and learning rate schedule during OLMo-2 7B model pretraining. **(C, D)** RankMe and $\alpha$ReQ, respectively, for OLMo-2 7B model vary non-monotonically across pretraining, demonstrating three key phases: "warmup", "entropy-seeking", and "compression-seeking". **(E, F)** Same as C,D, but for Pythia models, demonstrating the consistent existence of the three phases across model families and scales.

## 3.1 Phases of pretraining: Non-monotonic changes in representation geometry

During the LLM pretraining stage, standard metrics used for identifying optimization instabilities, e.g. loss or gradient norms, often decrease near-monotonically. While useful to practitioners while determining successful recipes for pretraining large models, these metrics carry limited information about the model capabilities and downstream behavior. We demonstrate, on the contrary, that the high-dimensional representation geometry metrics undergo non-monotonic changes. (And, later we demonstrate that these changes correlate with downstream performance).

Figure 2 illustrates this contrasting trend between the optimization metrics and representation geometry metrics during the pretraining of aforementioned family of LLMs. Specifically, we measured the RankMe [Garrido et al., 2023] and $\alpha$ReQ [Agrawal et al., 2022] metrics on the LLM's last layer representation of the last token while processing sequences from the FineWeb dataset [Penedo et al., 2024], and observed that there exist three distinct phases during the pretraining stage. Initially, there is a "warmup" phase, coinciding with the learning rate ramp-up, exhibiting a rapid collapse of the representations along the dominant data manifold directions. This relatively short phase is followed by an "entropy-seeking" phase characterized by a manifold expansion in several directions, and then a "compression-seeking" phase that imposes an anisotropic consolidation of the representation space along its principal eigenvectors. We observe these phases in both OLMo2 and Pythia family of models across different model sizes, indicating the consistent nature of non-monotonic changes in representation geometry during pretraining. It is worth noting that there could be emergence of additional "entropy-seeking" and "compression-seeking" with more pretraining, as in later stages of OLMo-2 7B model pretraining (c.f. Figure 2C). Notably, these phases persist even in smaller models below 1B parameters (Appendix Fig. 11), demonstrating the fundamental nature of this geometric evolution. Furthermore, these three-phase dynamics are consistently observed across intermediate layers throughout the network depth (Appendix Figure 7), confirming that the geometric evolution is not confined to just the final layer representations.

> **Key takeaway.** Despite near-monotonic loss, representation geometry exhibits a consistent, non-monotonic phase sequence ("warmup"; "entropy-seeking"; "compression-seeking"). These trends are stable across: (i) sample count $M$ and sequence length $L$; (ii) dataset choice within family; and (iii) layers (with last-layer sufficing for tracking), for both OLMo and Pythia.

## 3.2 Memorization & beyond: Distributional memorization happens in entropy-seeking phase

In this section, we seek to associate the different geometric phases to specific LLM behaviors. Downstream tasks that test the LLM's factual reasoning and language understanding abilities seem to improve with more pretraining. However, it is unclear to what extent this increase is due to an improvement in the model's memorization ability, i.e. how good is the model in "regurgitating" short-context phrases from the pretraining dataset, as opposed to a general language understanding,

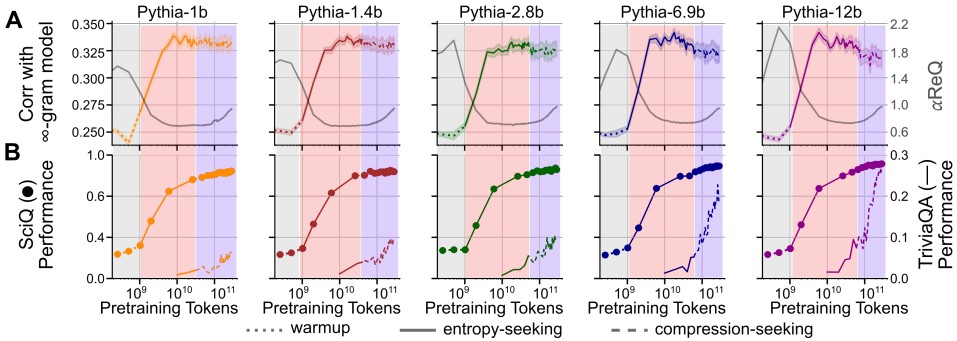

Figure 3: **Distinct learning phases are linked to different LLM capabilities. (A)** Memorization metric, i.e. spearman correlation between LLM and $\infty$-gram outputs, and representation geometry metric, $\alpha$ReQ, across Pythia models' (1–12B parameters) pretraining. Memorization peaks late in the "entropy-seeking" phase before plateauing or degrading slightly in the "compression-seeking" phase, suggesting that the former prioritizes capturing short-context n-gram statistics. **(B)** 0-shot performance on multiple-choice (SciQ) and factual question-answering (TriviaQA) tasks across pretraining. While accuracy on SciQ benefits from learning in both phases, accuracy on TriviaQA *groks* once the model learns long-context statistics, primarily in the "compression-seeking" phase.

i.e. leveraging long-context dependencies to generate reasonable output. We disentangle these two factors by using the distributional memorization metric Wang et al. [2025] presented in eq. (2) for Pythia models when processing sequences from the TriviaQA dataset [Joshi et al., 2017]. Notably, the $\infty$-gram model predominantly utilizes short- to medium-length suffixes (Appendix Table 7), making it an ideal baseline for measuring short-context memorization capabilities.

Figure 3 illustrates the memorization metric and task performance over the course of pretraining for Pythia models of 5 different sizes – ranging from 1B to 12B. Across all models, the distributional memorization metric increased during the "entropy-seeking" phase and peaked towards the end of this phase. Intuitively, this result suggests that the "entropy-seeking" phase is particularly important for learning short-context statistics, e.g. high-frequency n-grams, present in the pretraining corpus. This intuition is also supported by Wang et al. [2025] (c.f. Fig 12). Following this peak in the memorization metric, it plateaued (or slightly decreased) during the "compression-seeking" phase, suggesting that the model's output in this phase is guided by factors beyond n-gram statistics. Notably, the 0-shot accuracy on multiple-choice question-answering tasks, e.g. SciQ [Welbl et al., 2017], consistently improved throughout both the "entropy-seeking" and "compression-seeking" phases, potentially benefiting from both short- and long-context information learned in the respective phases.

However, 0-shot performance on factual question-answering tasks, e.g. TriviaQA [Joshi et al., 2017], demonstrate a sudden and dramatic rise in accuracy closely aligned with the saturation of the memorization metric. Consequently, most of the improvement in task accuracy happens during the "compression-seeking" phase, potentially benefiting from the long-context statistics learned in this phase, which are crucial for this task. The requirement of long-context dependencies for TriviaQA (and not SciQ) [Lou et al., 2024] seems to be – at least in part – responsible for the distinct performance on these tasks in the "entropy-seeking" and "compression-seeking" phases. Taken together, these findings outline a distinct association between each phase and the emergence of different LLM capabilities: short-context n-gram modeling during the "entropy-seeking" phase and long-context information aggregation during the "compression-seeking" phase.

> **Key takeaway.** "entropy-seeking" expands utilized dimensions (`RankMe` $\uparrow$, $\alpha$ReQ $\downarrow$), aligning with increased alignment to $\infty$-gram statistics (short-context distributional memorization). In contrast, "compression-seeking" anisotropically concentrates information (`RankMe` $\downarrow$, $\alpha$ReQ $\uparrow$), aligning with improvement in long-context QA accuracy even as memorization saturates.

### 3.3 Role of learning objective and optimization in learning dynamics

Having demonstrated the existence and salience of distinct learning phases, we now seek to understand the role of loss and optimization frameworks used in LLM pretraining in engendering these phases. Specifically, we studied the gradient descent dynamics while optimizing the cross-entropy loss in

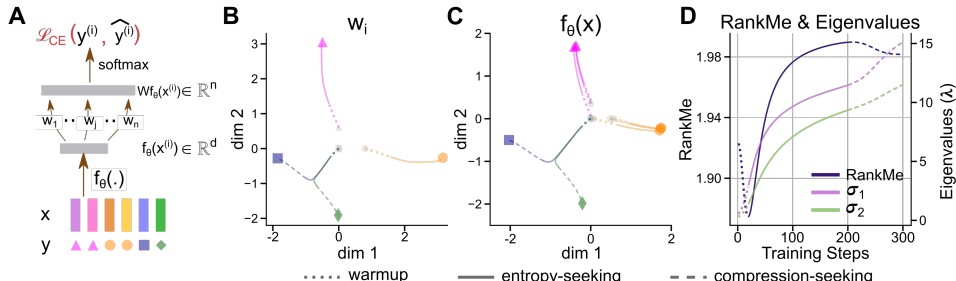

Figure 4: **Learning dynamics of cross-entropy loss replicate multiphase learning dynamics.**
**(A)** Schematic of a model with feature extractor $f_\theta(\in \mathbb{R}^d)$, linear classifier $W(\in \mathbb{R}^{n \times d})$ and
cross-entropy loss $\mathcal{L}_{CE}$. Skewed class distribution and information bottleneck ($d < n$) are critical
to replicate all three phases observed in LLM pretraining. **(B, C)** Classifier weights ($W_i$) and
feature representations ($f_\theta(x)$) demonstrate distinctive trajectories analogous to "warmup" (dotted),
"entropy-seeking" (solid), and "compression-seeking" (dashed) phases. **(D)** Quantitative spectral
metrics RankMe and eigenvalues, $\sigma_1, \sigma_2$.

an analytically-tractable setting — the model $f_\theta(x)$ is linear, i.e. $f_\theta(x) = \theta x \in \mathcal{R}^d$, and logits
are obtained (like in LLM models) as $z = W f_\theta(x) = W\theta x \in \mathcal{R}^{|\mathcal{V}|}$. The outputs are obtained by
applying a softmax operation on $z$ (see Figure 4A). We extended the results of Pezeshki et al. [2021]
to study how $W$ and $f_\theta(.)$ change when optimizing the loss using gradient descent. Notably, we found
two key properties of gradient descent that contribute to the emergent geometric properties of the
representation space (see Appendix B for formal statements):

- **Primacy bias**: Representations and weights corresponding to high-frequency tokens are
  learned earlier in training, compared to low-frequency tokens.
- **Selection bias**: Dominant directions in the representation space are more likely to be used
  for encoding new information, i.e. $\Delta\sigma_i \propto \sigma_i$

We demonstrate (c.f. Figure 4) that two conditions are necessary (see supplementary for controls)
for replicating the multiphase learning dynamics in our toy-model, as observed within LLMs: (1)
non-uniform class distribution, i.e. some tokens (or classes) occur more frequently than others in
the training data, and (2) information bottleneck, i.e. number of feature dimensions ($d$) is less than
the vocabulary size ($|\mathcal{V}|$). Note that these two conditions are common in LLM pretraining setups.
We defer the reader to Appendix Figures 12 and 13 for control experiments that ablate each of these
conditions and demonstrate a monotonic, saturating expansion of the representation space.

In the analytically tractable setup that satisfies the above conditions, we found that $f_\theta(.)$ and $W$ for
frequently-occurring classes are separated during the initial "warmup" phase (Figure 4B & C, dotted
lines). The corresponding eigenvectors of the weight and feature spaces also become aligned during
this phase. Following this initial eigenvector-alignment phase, there is an overall expansion in the
representation space that leads to higher confidence predictions for frequently-occurring classes.
This phase of volume expansion in the $f_\theta(.)$ and $W$ spaces is associated with an increasing effective
rank, akin to the "entropy-seeking" phase (Figure 4B & C, solid lines). Following this phase, the
infrequently-occurring classes start to separate into their own clusters in both spaces (Figure 4B & C,
dashed lines). Constrained by the information bottleneck condition, the system resorts to reusing the
feature space eigenvectors and more information is selectively encoded in the dominant direction (note
$\sigma_1$ grows faster compared to $\sigma_2$ after 200 steps in Figure 4D). This phase of anisotropic information
encoding leads to a reduction in RankMe, akin to the "compression-seeking" phase. Taken together,
these results suggest that gradient-based cross-entropy optimization dynamics under specific training
conditions may result in non-monotonic changes in representation geometry we observed in LLMs.

> **Key takeaway.** Gradient descent on cross-entropy with (i) skewed token frequencies and
> (ii) a representation bottleneck ($d \ll |\mathcal{V}|$) suffices to produce expansion $\rightarrow$ compression via
> eigenvector alignment and singular-value growth proportional to magnitude.

These mechanistic insights from simplified models establish fundamental principles governing
representation geometry evolution. We now turn to examining how these geometric transformations
manifest during post-training stages, where different loss functions further sculpt the representations.

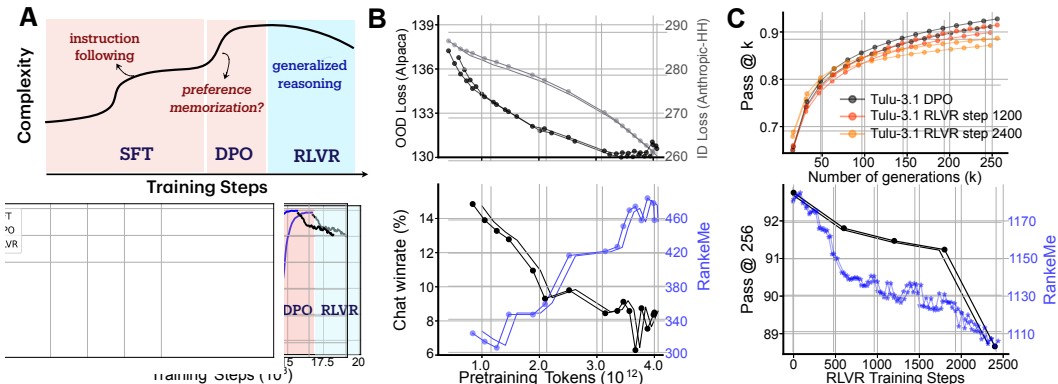

Figure 5: **Post-training induces distinct geometric transformations in model representations, aligned with specific behavioral changes. (A)** Conceptual overview of post-training (SFT, DPO and RLVR) **(top)**, corresponding $RankMe$ metrics from intermediate checkpoints of Llama-3.1-Tülu-3.1-8B **(bottom)** highlighting distinct progression for each stage. **(B)** Impact of pretraining on OLMo-2-1B SFT (Anthropic-HH): **(top)** longer pretraining improves in-distribution (ID) performance, while out-of-distribution (OOD) generalization (Alpaca farm) saturates **(bottom)** *Overtrained* models with higher RankMe exhibit markedly distinct outputs on AlpacaEval after undergoing SFT on two different datasets (Anthropic-HH and Alpaca farm). **(C)** RLVR post-training narrows base model's (Llama-3.1-8B-Tülu-3-DPO) exploratory behavior on AMC-23 (particularly at higher sampling counts e.g. $k = 256$), suggesting higher effective-rank facilitates better search.

## 3.4 Representation geometric changes during Post-Training stages

While pretraining establishes the initial structure of LLM representations, subsequent post-training is instrumental for refining model capabilities and aligning them with downstream objectives. Here, we investigate the geometric changes that occur during each post-training stage. Our analysis centers on the Tülu-3.1 models [Wang et al., 2024], which utilize a sequential three-stage post-training recipe — Supervised Fine-tuning (SFT), Direct Preference Optimization (DPO), and Reinforcement Learning with Verifiable Rewards (RLVR) applied to the LLaMA-3.1-8B [Grattafiori et al., 2024] base model.

**SFT exhibits "entropy-seeking" :** We find that SFT is associated with a monotonic increase in the RankMe, indicating an increase in the underlying representation manifold complexity. We hypothesize that the manifold expansion is related to instruction memorization on in-distribution (ID) examples, while reducing robustness to out-of-distribution (OOD) samples. To test this, we perform SFT with Anthropic-HH dataset on OLMo2-1B intermediate checkpoints. As shown in Figure 5 B, we find that with more pretraining the ID loss on Anthropic-HH improves monotonically, while the OOD loss (on Alpaca farm data) increases (see detailed ID/OOD loss in Appendix Figure 14). To understand the role of base-model geometry on the generalization gap, we perform SFT on Anthropic-HH (AH) and Alpaca farm (AF) datasets across checkpoints of OLMo2-1B, and measure chat winrates for AH using AF as reference on the AlpacaEval dataset. Strikingly, we find ( Figure 5B bottom) that while more pretraining coincides with an increase in RankMe, the winrates decrease for AH. Notably, a drop in winrate from 14% to 9% suggests that the LLM judge is better able to distinguish between the outputs of the two instruction-tuned models. This reinforces that "overtrained" base models, exhibiting higher RankMe, are more sensitive to distribution shifts under SFT [Springer et al., 2025].

**DPO exhibits "entropy-seeking" :** Prior works in self-supervised vision pretraining [Zhai et al., 2024, Ghosh et al., 2024] have established that contrastive learning objectives, e.g. SimCLR, are associated with an increase in representation complexity, as the network progressively learns the relevant eigenfunctions [Simon et al., 2023] to separate the positive and negative examples. We observe a similar trend in the DPO stage, notably a monotonic increase (decrease) in the RankMe ($\alpha$ReQ), c.f. fig. 5A. This parallel between the two settings can be attributed to the analogous formulations in the objective function. Note below that eq. (3) can be written as the Noise Contrastive Estimation (NCE) loss [Gutmann and Hyvärinen, 2010], often used in contrastive vision and multimodal pretraining [Oord et al., 2018, Chen et al., 2020, Radford et al., 2021], with *one* negative example.

$$\mathcal{L}_{DPO} = -\mathbb{E}_{x,y_w,y_l}\left[\log(\sigma(\hat{r}_\theta(x,y_w) - \hat{r}_\theta(x,y_l)))\right] = -\mathbb{E}_{x,y_w,y_l}\left[\log\frac{e^{\hat{r}_\theta(x,y_w)}}{e^{\hat{r}_\theta(x,y_w)} + e^{\hat{r}_\theta(x,y_l)}}\right] \quad (5)$$

**RLVR exhibits "compression-seeking" :** In sharp contrast to SFT and DPO, we observe that RLVR is associated with a monotonic decrease in `RankMe` (cf. Figure 5A). To probe the implications of this "compression-seeking" stage, we evaluate the unbiased `pass@k` performance on AMC-23 math benchmark. Figure 5C shows that while RLVR-training for 2400 steps outperforms the base (post-DPO) model at `pass@16`, the base model as well as an intermediate checkpoints outperform the RLVR-trained model at `pass@256`. This decline in `pass@256` performance as training progresses, reinforces prior work [Yue et al., 2025] suggesting that RLVR constraints the exploratory behavior of base models while amplifying some pre-existing behaviors of the base model [Zhao et al., 2025].

> **Key takeaway.** SFT/DPO (`RankMe` $\uparrow$, $\alpha$ReQ $\downarrow$) enhance in-distribution fit but increase sensitivity to dataset idiosyncrasies; RLVR (`RankMe` $\downarrow$, $\alpha$ReQ $\uparrow$) consolidates reward-aligned behaviors and narrows high-$k$ exploration (pass@$k$), consistent with reduced solution diversity.

## 4    Related Work

**Dynamics of Knowledge Acquisition and Representation Learning** A central theme in understanding neural networks is that learning is a dynamic, often phased process rather than a monolithic one. Recent work by Zucchet et al. [2025] identified distinct stages in how LLMs learn factual information, highlighting the formation of critical internal structures like attention circuits during performance plateaus. This notion of staged learning is further supported by the "Distributional Simplicity Bias" (DSB) established by Refinetti et al. [2023], Belrose et al. [2024], which posits that networks learn simpler statistical properties of data (e.g., lower-order moments) before more complex ones. Our work provides a geometric lens on these phenomena, using spectral measures to track how the effective dimensionality of representations evolve non-monotonically. Furthermore, Michaud et al. [2023] proposed that scaling laws and emergent abilities arise from learning discrete "quanta" of knowledge. DeMoss et al. [2024] explained grokking [Power et al., 2022] as a transition from high-complexity memorization to low-complexity generalization, measured via algorithmic information theory. Our spectral geometric phases offer a complementary perspective that could underpin these observed emergent jumps in performance and the dynamics of grokking.

**Post-Training Alignment and Reasoning** The adaptation of pretrained LLMs through fine-tuning is critical for aligning them with specific tasks and user preferences. Ren and Sutherland [2024] provided an empirical-NTK based framework to decompose the influence of fine-tuning updates, explaining complex behaviors such as hallucination amplification in SFT and the "squeezing effect" in DPO, where confidence in desired outputs can paradoxically decrease. Concurrently, Springer et al. [2025] identified "catastrophic overtraining," showing that excessive pretraining can make models overly sensitive to parameter changes, thereby degrading performance after SFT. Our work contributes to this area by demonstrating that different post-training strategies (SFT, DPO, RLVR) induce distinct transformations in the geometry and it's influence model capabilities.

## 5    Discussions

**Geometry of Pretraining: Memorization vs Generalization.** We show that LLM pretraining is multiphasic rather than monotonic, characterized mainly by "entropy-seeking" and "compression-seeking" phases. The observed geometric phases provide a quantitative framework for examining the relationship between memorizing short-context statistics and generalizing long-context information. The "entropy-seeking" phase expands the representational space to capture various short-context patterns, including n-gram memorization. Conversely, the "compression-seeking" phase promotes a more structured manifold and is likely to incentivize generalizable long-range language understanding. This geometric refinement process is consistent with and may offer an explanation for phenomena like *grokking*, where generalization capabilities can emerge after an initial period of fitting.

Our preliminary analysis further reveals the importance of full-spectrum information for model performance. When we ablate eigenvectors to retain only the top-k principal components, SciQ accuracy degrades dramatically (Appendix Table 8). For instance, retaining only the top 10 eigen-directions reduces Pythia-1B's accuracy from 0.838 to 0.225, while OLMo-2-7B drops from 0.970 to 0.155. Interestingly, removing the top eigen-directions has minimal impact, suggesting that information is distributed across the full spectrum rather than concentrated solely in dominant directions. This finding validates our use of full-spectrum metrics like `RankMe` and $\alpha$ReQ rather than top-k proxies, and underscores that effective language understanding requires the entire representational mani-

fold—not just its principal components. The necessity of preserving full spectral information aligns with the "compression-seeking" phase's anisotropic consolidation, which selectively strengthens certain directions while maintaining distributed representations across the manifold.

**Geometry of Post-Training: Alignment vs Exploration.** Different post-training recipes induce distinct shifts in LLM representation geometry, potentially explaining the model's behavioral changes. Supervised Fine-Tuning (SFT) typically drives an "entropy-seeking" dynamic, expanding the representational manifold for specific instruction-response examples. This manifold expansion can be seen as evidence for lazy-regime learning [Ren and Sutherland, 2024] during SFT, and points to a near-diagonal empirical NTK that results in an instance-level learning dynamics. Consequently, this dynamic improves in-distribution performance but risks overfitting due to higher representational capacity. In contrast, Reinforcement Learning from Verifiable Rewards (RLVR) promotes a "compression-seeking" dynamic, refining representations towards reward-aligned directions. This geometric compression may explain how RLVR amplifies and refines existing capabilities [Zhao et al., 2025], potentially by constraining representations to a more structured subspace while reducing its exploration ability [Yue et al., 2025]. In summary, SFT/DPO-induced rank expansion may foster preference memorization and exploratory behavior, while RLVR-induced consolidation amplifies model-capabilties towards reward-oriented, less diverse generation (c.f. Figure 5C).

**Limitations and Future Work** Tracing a model's geometry, whether "entropy-seeking" or "compression-seeking" , could inform more effective interventions for LLM development and evaluation, such as the selection of optimal pretraining checkpoints for targeted fine-tuning or designing training strategies that deliberately navigate these geometric phases. Our findings have several limitations: (i) computational constraints limited our analysis to models up to 12B parameters, though the phases persist across scales from 160M to 12B; (ii) spectral metric computation requires $\sim$10K samples and scales quadratically with hidden dimension (iii) our theoretical analysis assumes simplified linear feature extractors, leaving the extension to full transformer architectures as future work; (iv) we focused on English-language models trained with standard objectives, and whether similar phases emerge in multilingual or alternatively-trained models remains unexplored. Furthermore, our findings are primarily correlational; establishing causal connections between geometric dynamics and emergent capabilities requires additional investigation.

# 6 Conclusion

We show that LLMs undergo non-monotonic representation geometry changes, often masked by steadily decreasing training loss. By employing spectral metrics of feature covariates (`RankMe` and $\alpha$ReQ), we delineate three distinct pretraining phases: "warmup" , "entropy-seeking" (correlating with n-gram memorization), and "compression-seeking" (correlating with long-context generalization). We further demonstrate that post-training recipes induce specific geometric changes: SFT/DPO exhibit "entropy-seeking" dynamics, whereas RLVR exhibit "compression-seeking" dynamics. These results provide a quantitative framework for guiding future advancements in LLM development.

**Impact Statement** The goal of our work is to advance the understanding of internal representations of LLMs. Although there are potential downstream societal consequences of this technology, we feel there are no direct consequences that must be specifically highlighted here.

# Acknowledgments

The authors would like to thank Koustuv Sinha for insightful discussions that helped shape the scope of the project and Jacob Mitchell Springer for helping setup the OLMo-2 supervised finetuning pipeline. We are grateful to the OLMo team, particularly Nathan Lambert, Dirk Groeneveld, and Bailey Kuehl, for providing access to the OLMo-2 checkpoints (especially OLMo-2-1B) that enabled this research. The authors are also grateful to Daniel Levenstein, Johannes von Oswald, Jonathan Cornford, Mandana Samiei, Tejas Vaidhya, and Zahraa Chorghay for their comments and feedback. A.G. was supported by Vanier Canada Graduate Scholarship. G.L. was supported by NSERC (Discovery Grant RGPIN2018-04821), the Canada Research Chair in Neural Computations and Interfacing, CIFAR (Canada AI Chair), as well as IVADO and the Canada First Research Excellence Fund. B.A.R. was supported by NSERC (Discovery Grant: RGPIN-2020-05105; Discovery Accelerator Supplement: RGPAS-2020-00031) and CIFAR (Canada AI Chair; Learning in Machines and Brains Fellowship).

The authors also acknowledge the material support of NVIDIA in the form of computational resources, as well as the compute resources, software and technical help provided by Mila (mila.quebec).

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

# A  Model and Dataset Details

## A.1  Model Architecture and training configurations

Table 1: Comparison of model architectures and training setups.

|  | Pythia | OLMo-2 |
|---|---|---|
| **Position Embedding** | Learned | Rotary (RoPE) |
| **Norm Type** | LayerNorm | RMSNorm |
| **Norm Position** | Pre-layer | Pre-layer |
| **Dataset** | The Pile (825 GB) | OLMoStack (4T tokens) |
| **Optimizer** | AdamW | AdamW |
| **LR Scheduler** | Cosine decay | Linear decay w/ warmup |
| **Loss Function** | Cross-Entropy | Cross-Entropy |

Table 2: Tülu Model Architecture and Training Setup

| Component | Tülu |
|---|---|
| **Base Models** | Llama 3 base models |
| **Position Embedding** | Inherited from base model |
| **Normalization Type** | Inherited from base model (LayerNorm) |
| **Normalization Position** | Pre-layer |
| **Instruction Datasets** | Tulu3 Mixture(FLAN V2, OpenAssistant, WildChat GPT-4) |
| **Training Techniques** | SFT, DPO, RLVR |
| **Optimizer** | AdamW |
| **Learning Rate Scheduler** | Linear decay with warmup |
| **Loss Function** | Cross-Entropy |

## A.2  Dataset Details

In this section, we provide an overview of the datasets used in our experiments.

**FineWeb**: The FineWeb dataset [Penedo et al., 2024] consists of more than 15T tokens of cleaned and deduplicated english text obtained from the web using CommonCrawl. While the full dataset contains 15T tokens, we use the smallest subset, i.e. a subsampled version of the dataset consisting of 10B tokens. The dataset is accessible on HuggingFace at https://huggingface.co/datasets/HuggingFaceFW/fineweb.

**WikiText**: The Wikitext dataset [Merity et al., 2016] is a collection of over 100 million tokens extracted from the set of verified Good and Featured articles on Wikipedia. We use only a subset of the dataset to perform early evaluations of `RankMe` and $\alpha$ReQ, before running our final experiments using FineWeb.

**SciQ**: The SciQ dataset [Welbl et al., 2017] contains over 13K crowdsourced science exam questions about physics, chemistry and biology, among many others. The questions are in multiple-choice format with 4 answer options each. The dataset is accessible on HuggingFace at https://huggingface.co/datasets/allenai/sciq.

**TriviaQA**: The TriviaQA dataset [Joshi et al., 2017] is a reading comprehension dataset containing over 650K question-answer-evidence triples. We use the TriviaQA dataset to evaluate a model's ability to incorporate long-context information from the question in order to correctly answer it. The dataset is accessible on HuggingFace at https://huggingface.co/datasets/mandarjoshi/trivia_qa.

**LAMBADA OpenAI**: This dataset [Radford et al., 2019] is comprised of the LAMBADA test split, pre-processed by OpenAI, and contains machine translated versions of the split in German, Spanish, French and Italian. We use this dataset to evaluate the model's text understanding capabilities. The dataset is accessible on HuggingFace at https://huggingface.co/datasets/EleutherAI/lambada_openai.

**Anthropic Helpful-Harmless (HH)**: The Anthropic-HH dataset provides human preference data about helpfulness and harmlessness, and is meant to be used for training preference models in a Reinforcement Learning with Human Feedback (RLHF) setting. However, we use a variant of

this dataset for SFT. Specifically, we generate a human-assistant chat dataset of $\sim 161K$ samples by parsing the "chosen" responses for each instruction from the original dataset and using it to finetune a base model by treating the "chosen" response as the target (similar to Springer et al. [2025]). While such a use of this dataset is discouraged in practical settings, we use this modified dataset as a testbed for our SFT experiments. The original dataset is accessible on HuggingFace at https://huggingface.co/datasets/Anthropic/hh-rlhf.

**AlpacaFarm Human-ANN chat (AlpacaFarm)**: This dataset is created by following a similar procedure as mentioned above for the Anthropic-HH dataset, but for the Human Evaluation dataset of the AlpacaFarm evaluation set [Dubois et al., 2023]. As a result, this dataset consists of $\sim 17.7K$ samples, and is used as a positive control in our SFT experiments. Models that are finetuned on this dataset are expected to perform well on the AlpacaEval chat task (see below), compared to models that are finetuned on a different dataset. This positive control is essential to disentangle the in-distribution vs out-of-distribution abilities of a SFT-model. The original dataset is accessible on HuggingFace at https://huggingface.co/datasets/tatsu-lab/alpaca_farm.

**AlpacaEval**: AlpacaEval is an LLM-based automatic evaluation setup for comparing chat models in a fast, cheap and replicable setting. We use AlpacaEval as a test bench to study the behavior of models after undergoing SFT. Models that are finetuned on the AlpacaFarm dataset are expected to produce better chat models and generate responses more aligned to human-preferred responses to instructions in the AlpacaEval setup. We defer the reader to the corresponding github repository for further details of the evaluation setup.

**AMC23**: The AMC23 benchmark refers to a specific set of evaluations based on the American Mathematics Competitions. This benchmark is designed to assess the mathematical reasoning capabilities of advanced AI models using problems characteristic of the AMC series. For the evaluation of AMC23, we utilize the resources and methodologies found in the Qwen2.5-Math repository. This repository is accessible at `https://github.com/QwenLM/Qwen2.5-Math` and provides the framework for our assessment process.

### A.3 Compute and hyperparameter configuration details

**Compute resources:** All of our LLM inference experiments were run either on a single 80GB A100 or a 40GB L40S GPU. The finetuning experiments (SFT and DPO) were run on a single node consisting of 4 A100 GPUs.

**Codebase:** All code and data will be released at `https://github.com/melodylizx/Tracing-the-representation-geometry-of-language-models` .

| Hyperparameter | Value |
|---|---|
| Dataset | FineWeb sample-10BT |
| Max sequence length | 512 |
| Number of sequences | 15000 |
| Batch size | 16 |

Table 3: Hyperparameter configurations used for computing `RankMe` and $\alpha$ReQ in Figure 2.

### A.4 Reproducing Tülu-3-8B SFT and DPO

We follow instructions from [https://github.com/allenai/open-instruct] for reproducing and gathering the intermediate stage checkpoints (both for SFT and DPO) without changing any hyperparamters.

| Hyperparameter | Value |
|---|---|
| SFT dataset | Anthropic-HH or AlpacaFarm Human-ANN chat (train split) |
| Max sequence length | 4096 |
| Batch size | 16 |
| Gradient accumulation steps | 16 |
| Learning rate | 1e-5 |
| Learning rate schedule | Linear decay with 10% warmup |
| Number of epochs | 2 |
| Loss reduction | sum |
| Seeds | 0, 7, 8, 42, 420 |

Table 4: Hyperparameter configurations used for Supervised FineTuning (SFT).

| Hyperparameter | Value |
|---|---|
| Base model | OLMo2-1B |
| In-distribution dataset | Anthropic-HH (test split) |
| Out-of-distribution dataset | AlpacaFarm Human-ANN chat (train split) |
| Max sequence length | 1024 |
| Number of sequences | 10000 |
| Batch size | 32 |

Table 5: Hyperparameter configurations used for ID and OOD loss eval.

| Hyperparameter | Value |
|---|---|
| Base model | OLMo2-1B |
| Dataset | AlpacaEval (test split) |
| Max new tokens | 1024 |
| LM judge | Cohere Command A |

Table 6: Hyperparameter configurations used for chat winrate on AlpacaEval.

# B  Gradient Descent and Cross-entropy theory

## B.1  Setup

Let $s$ denote an individual instance (or input token sequence of language), with its true class identity (or the next token's index in vocabulary) given by $y(s)$. An (LLM) encoder, parameterized by $\theta$, processes $s$ to produce its contextualized embedding, $f_\theta(s) \in \mathbb{R}^d$, where $d$ is the embedding dimension. For a batch, $S$, of $b$ such instances, the encoder outputs a matrix $f_\theta(S) \in \mathbb{R}^{b \times d}$. Subsequently, predictions $\hat{\mathbf{y}} \in \mathbb{R}^{b \times |\mathcal{V}|}$ are generated by multiplying this batch embedding with a weight matrix $W \in \mathbb{R}^{d \times |\mathcal{V}|}$, where $|\mathcal{V}|$ is the vocabulary size (or number of classes):

$$\hat{\mathbf{y}} = f_\theta(S)W$$

To simplify the setting such that it is analytically-tractable, we assume the embedding function $f_\theta(s)$ to be modeled as a linear transformation of the input, i.e. $f_\theta(s) = \theta^T s$, where $\theta \in \mathbb{R}^{d_{in} \times d}$ is a parameter matrix. For a batch, of $b$ such instances, represented as a matrix $S \in \mathbb{R}^{b \times d_{in}}$ whose row vectors are orthonormal (i.e., $SS^T = \mathbf{I}$). The batch embedding is, therefore, $f_\theta(S) = S\theta \in \mathbb{R}^{b \times d}$. Here, $d_{in}$ is the input feature dimension and $d$ is the embedding dimension.

**Note:** By imposing this i.i.d. assumption, we ensure that learning on one sample does not change the output of another sample, i.e. no inter-sample interference. While we admit that this assumption is unrealistic, and learning to predict the next token of one sequence in an autoregressive setup affects the output of another sequence, we believe that this assumption enables a first step towards understanding the implicit effect of cross-entropy loss optimization using gradient descent. We note that our results do not strictly depend on this assumption, and can be extended to non-i.i.d. samples. We leave this to future work.

**Note 2:** Note that our assumption of a linear embedding model, $f_\theta$, is also an aberration from the transformer-based LLMs. However, we focus on the effect of loss and optimization in this section and leave further investigation into the implicit bias of architecture to future studies.

## B.2  Linear approximation of Cross-entropy loss: Legendre Transform

Let us start by defining the cross-entropy loss for one example, $s$, which belongs to class $c$ as:

$$\mathcal{L}_{CE}(s) = -\log\left(\frac{e^{\hat{y}_c}}{\sum_j e^{\hat{y}_j}}\right) = -\hat{y}_c + \log\left(\sum_j e^{\hat{y}_j}\right) \tag{6}$$

Note that eq. (6) is nonlinear in $\hat{\mathbf{y}}$, thereby making it harder to analyze the dynamical system in the parameter space that is imposed by gradient descent. In order to arrive at an analytical understanding of the gradient-induced dynamics when optimizing eq. (6), we will do a linear approximation of $\mathcal{L}_{CE}$ using Legendre Transform, similar to Pezeshki et al. [2021]. Specifically, we will derive the Legendre transform of the nonlinear term, $\log(\sum_j e^{\hat{y}_j})$.

Intuitively, we want to approximate eq. (6) such that it changes linearly with changes in $\hat{\mathbf{y}}$. The key motivation for using Legendre transform is to ignore the second (and higher) order effects of a "small" change in $\hat{\mathbf{y}}$ on $\mathcal{L}_{CE}$. Formally, we want the following:

$$\hat{\mathcal{L}}_{CE}(s) = -\hat{y}_c + \alpha^T \hat{\mathbf{y}} + g(\alpha) \tag{7}$$

where $\alpha$ is the slope of the nonlinear term at $\mathbf{x} = \hat{\mathbf{y}}$.

$$\alpha = \nabla_{\mathbf{x}} log\left(\sum_j e^{x_j}\right)\Big|_{\mathbf{x}=\hat{\mathbf{y}}}$$

$$\implies \alpha_i = \frac{\partial}{\partial x_i} \log\left(\sum_j e^{x_j}\right)\Big|_{x_j=\hat{y}_j} = \frac{e^{\hat{y}_i}}{\sum_j e^{\hat{y}_j}} \tag{8}$$

Note that: $\boxed{\sum_i \alpha_i = 1}$

To simplify things, let us denote $C = \sum_j e^{\hat{y}_j}$. Substituting this in eq. (8), $\hat{y}_i = log(\alpha_i) + log(C)$.

Now we need to find $g(\alpha)$ such that $f(\hat{\mathbf{y}}) = \alpha^T \hat{\mathbf{y}} + g(\alpha)$.

$$g(\alpha) = f(\hat{\mathbf{y}}) - \alpha^T \hat{\mathbf{y}} = log(\sum_j e^{\hat{y}_j}) - \alpha^t \hat{\mathbf{y}} = log(\sum_j e^{\hat{y}_j}) - \sum_j \alpha_j \hat{y}_j$$

$$= log(C) - \sum_j \alpha_j log(\alpha_j) - \sum_j \alpha_j log(C)$$

$$= log(C) - \sum_j \alpha_j log(\alpha_j) - log(C) \qquad [\text{Using} \sum_j \alpha_j = 1]$$

$$= -\sum_j \alpha_j log(\alpha_j) = H(\alpha) \tag{9}$$

where $H(\alpha)$ denotes the Shannon entropy of a probability distribution defined by $\alpha_i$'s.

Substituting eq. (9) in eq. (7), we get the expression for the linearized cross-entropy loss:

$$\hat{\mathcal{L}}_{CE}(s) = -\hat{y}_c + \alpha^T \hat{\mathbf{y}} + H(\alpha) \tag{10}$$

### B.3 Gradient descent dynamics of linearized cross-entropy loss

$$\hat{\mathcal{L}}_{CE}(s) = -\hat{y}_c + \alpha^T \hat{\mathbf{y}} + H(\alpha) = -\hat{y}_c + \sum_j \alpha_j \hat{y}_j + H(\alpha)$$

$$= -f_\theta(s)^T w_c + \sum_j \alpha_j f_\theta(s)^T w_j + H(\alpha)$$

$$\nabla_{f_\theta(s)} \hat{\mathcal{L}}_{CE}(s) = -w_c + \sum_j \alpha_j w_j = \sum_j (\alpha_j - \delta_{j=c}) w_j$$

$$\nabla_{w_i} \hat{\mathcal{L}}_{CE}(s) = -f_\theta(s) \delta_{i=c} + \alpha_i f_\theta(s) = (\alpha_i - \delta_{i=c}) f_\theta(s)$$

where $\delta_{(.)}$ is the Dirac-delta function, i.e. its value is 1 when the condition in subscript is true and 0 otherwise.

Denoting $\tilde{\alpha}_i = (\alpha_i - \delta_{i=c})$, we arrive at the gradient equations for $f_\theta(s)$ and $w_i$'s:

$$\nabla_{f_\theta(s)} \hat{\mathcal{L}}_{CE}(s) = \sum_j \tilde{\alpha}_j w_j \quad , \quad \nabla_{w_i} \hat{\mathcal{L}}_{CE}(s) = \tilde{\alpha}_i f_\theta(s) \tag{11}$$

We can easily extend eq. (11) to multiple examples $\{s_1, s_2 \cdots s_b\}$ and write the gradient descent update (using learning rate $\eta$) equations as:

$$\dot{f}_\theta(s_j) = -\eta \sum_i \tilde{\alpha}_i(s_j) w_i \quad , \quad \dot{w}_i = -\eta \sum_j \tilde{\alpha}_i(s_j) f_\theta(s_j)$$

$$\implies \dot{f}_\theta = -\eta A W^T \quad , \quad \dot{W} = -\eta f_\theta^T A \tag{12}$$

where

$$A_{ij} = \begin{cases} \alpha_j(s_i) - 1 & \text{if } c_i = j \\ \alpha_j(s_i) & \text{else} \end{cases} \quad (i^{th} \text{ example, } s_i, \text{ belongs to the class } j)$$

### B.4 A useful matrix algebra result

**Lemma 1.** *Let $W(t)$ be a time-varying matrix with singular value decomposition (SVD): $W(t) = U(t)S(t)V(t)^T$, where $U(t)$ and $V(t)$ are orthogonal matrices corresponding to the left and right singular vectors, respectively, and $S(t) = diag(\sigma_1(t), \sigma_2(t), \ldots, \sigma_k(t))$ contains the singular values along its diagonal. Let $u_k(t)$ and $v_k(t)$ denote the $k^{th}$ column vectors of $U(t)$ and $V(t)$, respectively. Then the time derivative of the $k^{th}$ singular value, $\sigma_k(t)$, is given by:*

$$\dot{\sigma}_k(t) = u_k(t)^T \dot{W}(t) v_k(t)$$

*Proof.* For sake of brevity, we will drop the explicit time-dependence of each matrix from the notations. Let us write the singular vector decomposition (SVD) of matrix, $W = USV^T$. Using the product rule of differentiation:

$$\dot{W} = \dot{U}SV^T + U\dot{S}V^T + US\dot{V}^T$$
$$\implies U^T\dot{W}V = U^T\dot{U}S + \dot{S} + S\dot{V}^TV$$
$$\implies \dot{S} = U^T\dot{W}V - U^T\dot{U}S - S\dot{V}^TV$$
$$\implies \dot{\sigma}_k = u_k^T\dot{W}v_k - u_k^T\dot{u}_k\sigma_k - \sigma_k\dot{v}_k^Tv_k \tag{13}$$

where the last line is the expression for the $k^{th}$ diagonal element of $S$. By definition of orthonormal vectors, $u_k^Tu_k = 1$. So, $\dot{u}_k^Tu_k + u_k^T\dot{u}_k = 0$. Since $\dot{u}_k^Tu_k$ is a scalar, $\dot{u}_k^Tu_k = u_k^T\dot{u}_k$. Therefore, $\dot{u}_k^Tu_k = 0$. Similarly, $\dot{v}_k^Tv_k = 0$. Therefore,

$$\dot{\sigma}_k = u_k^T\dot{W}v_k$$

$\square$

## B.5 Formal versions of theoretical results and proofs

**Theorem 1.** *Let $f_\theta = U_1S_1V_1^T$ and $W = U_2S_2V_2^T$ denote the respective singular value decompositions (SVDs) of non-degenerate matrices $f_\theta$ and $W$, respectively. If the system is initialized such that $f_\theta^T f_\theta = WW^T$, then it holds that:*

$$V_1 = U_2 \quad , \quad S_1^2 = S_2^2$$

*Proof.* Let us start from the learning dynamics imposed by gradient-descent:

$$\dot{f}_\theta = -\eta AW^T \quad , \quad \dot{W} = -\eta f_\theta^T A \tag{14}$$

Let us write $f_\theta$ and $W$ as their respective singular value decomposed form, i.e. say $f_\theta = U_1S_1V_1^T$ and $W = U_2S_2V_2$. Consider the dynamics of $f_\theta^T f_\theta$ and $WW^T$:

$$\frac{d}{dt}(f_\theta^T f_\theta) = \dot{f}_\theta^T f_\theta + f_\theta \dot{f}_\theta = (-\eta AW^T)^T f_\theta + f_\theta^T(-\eta AW^T)$$
$$= -\eta WA^T f_\theta - \eta f_\theta^T AW^T \tag{15}$$

$$\frac{d}{dt}(WW^T) = \dot{W}W^T + W\dot{W}^T = (-\eta f_\theta^T A)W^T + W(-\eta f_\theta^T A)^T$$
$$= -\eta f_\theta^T AW^T - \eta WA^T f_\theta \tag{16}$$

From eqs. (15) and (16), it is clear that $\frac{d}{dt}(f_\theta^T f_\theta) = \frac{d}{dt}(WW^T)$, i.e. $f_\theta^T f_\theta = WW^T + C$, for some constant $C$. If we assume the initialization to be such that $C = 0$ and $f_\theta$ and $W$ are non-degenerate, we have:

$$f_\theta^T f_\theta = WW^T \implies V_1S_1^2V_1^T = U_2S_2^2U_2^T$$

By uniqueness of SVD (for positive semi-definite matrices):

$$\boxed{\begin{aligned} V_1 = U_2 &\implies V_1^TU_2 = I \\ S_1^2 &= S_2^2 \end{aligned}}$$

$\square$

**Theorem 2.** *Let $f_\theta, W$ be the matrices whose dynamics are governed by the gradient-descent equations as previously defined. Given the conditions from Theorem 1, the magnitude of the time derivatives of the $i^{th}$ singular values of $f_\theta$ and $W$ are proportional to their respective singular values:*

$$\|\dot{\sigma}_{1i}\| \propto \sigma_{1i}$$
$$\|\dot{\sigma}_{2i}\| \propto \sigma_{2i}$$

*Furthermore, assuming uniform class prediction at initialization and that number of classes, $|\mathcal{V}| \gg 1$, the time derivatives are bounded by the dominant class size:*

$$\|\dot{\sigma}_{1i}\|, \|\dot{\sigma}_{2i}\| \propto \mathcal{O}(\mathcal{N}(c^{(0)}))$$

*where $\mathcal{N}(c^{(0)})$ denotes the number of instances belonging to the dominant class $c^{(0)}$.*

*Proof.* Let us start from the results of Theorem 1: $S_1^2 = S_2^2 \implies \sigma_{1i}^2 = \sigma_{2i}^2 \ \forall i$. So, $\sigma_{1i} = \pm\sigma_{2i}$. Using this relation, we can simplify the expression of $\sigma_{1i}$ dynamics. From Lemma 1,

$$\begin{aligned}
\dot{\sigma}_{1i} = u_{1i}^T \dot{f}_\theta v_{1i} &= -\eta u_{1i}^T A W^T v_{1i} \\
&= -\eta u_{1i}^T A (U_2 S_2 V_2^T)^T v_{1i} = -\eta u_{1i}^T A V_2 S_2 U_2^T v_{1i} \\
&= -\eta u_{1i}^T A V_2 S_2 V_1^T v_{1i} \qquad \text{[Using Theorem 1]} \\
&= -\eta \sum_j (u_{1i}^T A v_{2j}) \sigma_{2j} (v_{1j} v_{1i}) = -\eta \sum_j (u_{1i}^T A v_{2j}) \sigma_{2j} \delta_{i=j}
\end{aligned}$$

$$\implies \dot{\sigma}_{1i} = -\eta (u_{1i}^T A v_{2i}) \sigma_{2i} \tag{17}$$

Similarly, we can simplify the dynamics for $\sigma_{2i}$:

$$\dot{\sigma}_{2i} = -\eta (u_{1i}^T A v_{2i}) \sigma_{1i} \tag{18}$$

For sake of brevity, let us denote $(u_{1i}^T A v_{2i}) = g_i$. Using the relationship between $\sigma_{1i}$ and $\sigma_{2i}$, we can simplify eqs. (17) and (18) as:

$$\dot{\sigma}_{1i} = -\eta g_i(\pm\sigma_{1i}) = \mp\eta g_i \sigma_{1i} \quad , \quad \dot{\sigma}_{2i} = -\eta g_i(\pm\sigma_{2i}) = \mp\eta g_i \sigma_{2i} \tag{19}$$

$$\boxed{\implies \|\dot{\sigma}_{1i}\| \propto \sigma_{1i} \quad , \quad \|\dot{\sigma}_{2i}\| \propto \sigma_{2i}} \tag{20}$$

Also, note that $g_i = u_{1i}^T A v_{2i} = \sum_{j,k} u_{1ij} A_{jk} v_{2ik}$, where $A_{jk} = \{\alpha_k(s_j) - 1, \alpha_k(s_j)\}$. Therefore, $A_{jk} \in (-1, 1)$.

At initialization, WLOG $\alpha_k(s_j) \approx \frac{1}{|\mathcal{V}|} \ \forall j, k$, i.e. uniform class prediction. Additionally, assuming $|\mathcal{V}| >> 1$, we can estimate $g_i$ as the following:

$$g_i = \sum_{j,k} u_{1ij} A_{jk} v_{2ik} = \sum_k \left( \sum_{j \in \{c_j = k\}} u_{1ij}(\alpha_k(s_j) - 1) v_{2ik} + \sum_{j \in \{c_j \neq k\}} u_{1ij} \alpha_k(s_j) v_{2ik} \right)$$

$$\implies g_i \approx \sum_k \left( (\frac{1}{|\mathcal{V}|} - 1) \sum_{j \in \{c_j = k\}} u_{1ij} v_{2ik} + \frac{1}{|\mathcal{V}|} \sum_{j \in \{c_j \neq k\}} u_{1ij} v_{2ik} \right)$$

$$\approx -(\sum_k v_{2ik})(\sum_{j \in \{c_j = k\}} u_{1ij}) = \mathcal{O}(\mathcal{N}(c^0)) \tag{21}$$

where $c^{(0)}$ denotes the dominant class, i.e. the class with most number of instances. Combining eq. (21) with eq. (19), we get the desired result:

$$\boxed{\|\dot{\sigma}_{1i}\|, \|\dot{\sigma}_{2i}\| \propto \mathcal{O}(\mathcal{N}(c^0))} \tag{22}$$

$\square$

# C Additional Experimental results

## C.1 Generative behavior of LLMs: Qualitative inspection

| stage1-step1000-5B | stage1-step920000-3895B |
|---|---|
| **Sunlight filtered through dense canopy, dappling the forest floor** | |
| The first thing you can see is that the first thing you see in the first place is that you can see the first thing you see in the first place. The second thing you see in the first place is that you can see the first thing you see in the first place. The second thing you see in the second place is that you can see the first thing you see | The air was still and warm, and the only sound was the occasional chirp of a bird. The forest was quiet, and the only movement was the slow, steady sway of the trees.\n\nA small, furry creature scurried through the underbrush, its eyes darting from side to side. It was a squirrel, and it was looking for food. It had |
| **An old, weathered map lay spread across the wooden table.** | |
| The table was a large, large, and large, and the table was a large, large, and large, and the table was a large, and the table was a large, and the table was a large, and the table was a large, and the table was a large, and the table was a large, and the table was a large, and the table | The map was of the area around the town of Harken, and it was marked with a number of locations. The map was covered with notes, and the notes were covered with a number of different colored markers. The markers were of different shapes and sizes, and they were all labeled with names and numbers. The map was a mess, and it was obvious that it |
| **A lone ship sailed silently towards the crimson horizon** | |
| The sky was a dark, dark, dark, dark, dark, dark, dark, dark, dark, dark, dark, dark, dark, dark, dark, dark, dark, dark, dark, dark, dark, dark, dark, dark, dark, dark, dark, dark, dark, dark, dark, dark, dark, dark, dark, dark, dark, dark, dark | The ship was a small, two-masted vessel, with a single mast on the stern. The ship was painted a dark blue, with a white stripe running along the hull. The ship was a pirate ship, and the captain was a pirate. The captain was a pirate, and the ship was a pirate ship. The captain was a pirate, and the ship was |
| **Intricate carvings covered the surface of the ancient stone tablet** | |
| The first part of the 19th century was the first part of the 19th century. The first part of the 19th century was the first part of the 19th century. The first part of the 19th century was the first part of the 19th century. The first part of the 19th century was the first part of the | The carvings were so intricate that they were almost impossible to decipher. The tablet was covered in a thick layer of dust and dirt. The dust and dirt had been there for so long that it had become a part of the stone. It was as if the dust and dirt had been there since the beginning of time.\n\nThe tablet was covered in a thick layer of dust |
| **A faint melody drifted from an open window down the cobblestone street** | |
| The two cars are the most expensive and the most expensive and the most expensive. The cars are the most expensive and the most expensive and the most expensive and the most expensive and the most expensive and the most expensive and the most expensive and the most expensive | The music was soft and sweet, and it was enough to make the young man smile. He was a tall, handsome man with dark hair and eyes. He was dressed in a fine suit, and he carried a violin case in his hand. He walked with a confident stride, and he seemed to be in no hurry. He stopped in front of a small house and knocked |
| **An unspoken tension filled the crowded room, heavy and palpable** | |
| The room was a large, heavy, and very large, very large, very large, very large, very large, very large, very large, very large, very large, very large, very large, very large, very large, very large, very large, very large, very large, very large, very | The air was thick with the scent of fear and the stench of death. The room was silent, save for the sound of the wind howling outside. The only light came from the flickering flames of the fire. The room was dark and cold, and the only warmth came from the bodies of the men and women huddled together in the corner. The smell |
| **A single drop of rain landed on the dusty, parched earth** | |
| The earth was a great, strong, strong, strong, strong, strong, strong, strong, strong, strong, strong, strong, strong, strong, strong, strong, strong, strong, strong, strong, strong, strong, strong, strong, strong, strong, strong, strong, strong, strong, strong, strong, strong, strong, strong, strong | It was the first rain in months. The earth drank it up greedily, and the thirsty plants and trees drank it up too. The thirsty animals drank it up. The thirsty people drank it up. The rain was a blessing. It was a gift from God. It was a gift of life.\n\nThe rain continued to fall. The earth drank it up. The plants |

Figure 6: Evidence of echolalia: Generated text from early checkpoints (step-1000) of OLMo-2-7B models, compared to that from later stages of pretraining (step-920000).

## C.2 Examining Context Statistics and the Necessity of Full-Spectrum Information

| Suffix Length | Frequency (%) |
|---|---|
| $\leq 3$ | 25.41 |
| 4 | 46.61 |
| 5 | 16.71 |
| 6 | 6.08 |
| 7 | 2.40 |
| 8 | 1.46 |
| $> 8$ | 1.34 |

Table 7: $\infty$-**gram context in TriviaQA.** Suffix lengths reveal focus on short- to mid-context statistics.

| Model | Original | Top-10 | | Top-50 | |
|---|---|---|---|---|---|
| | | Removed | Retained | Removed | Retained |
| Pythia-1B | 0.838 | 0.849 | 0.225 | 0.835 | 0.318 |
| Pythia-1.4B | 0.866 | 0.855 | 0.232 | 0.859 | 0.324 |
| Pythia-2.8B | 0.884 | 0.880 | 0.219 | 0.873 | 0.317 |
| Pythia-6.9B | 0.896 | 0.893 | 0.202 | 0.906 | 0.327 |
| OLMo-2-1B | 0.953 | 0.943 | 0.199 | 0.954 | 0.326 |
| OLMo-2-7B | 0.970 | 0.966 | 0.155 | 0.970 | 0.308 |

Table 8: **Full-spectrum information is required.** Retaining only top eigen-directions markedly degrades SciQ accuracy.

### C.3  Computing spectral metrics, `RankMe` and $\alpha$**ReQ**

| Model | $\alpha$**ReQ (p-value)** | `RankMe` **(p-value)** |
|---|---|---|
| Pythia-1B | 0.810 (1.50e-5) | -0.759 (1.04e-4) |
| Pythia-1.4B | 0.668 (1.29e-3) | -0.713 (4.18e-4) |
| Pythia-2.8B | 0.694 (6.88e-4) | -0.635 (2.63e-3) |
| Pythia-6.9B | 0.837 (4.20e-6) | -0.885 (2.19e-7) |
| Pythia-12B | 0.836 (4.42e-6) | -0.839 (3.79e-6) |
| OLMo2-1B | 0.540 (4.920e-7) | -0.616 (3.201e-9) |

Table 9: **SciQ accuracy correlates with spectral geometry.** Positive correlation with $\alpha$ReQ (compactness) and negative correlation with `RankMe` (effective dimensionality) across Pythia (1–12B) and OLMo2-1B models. The p-values in parentheses indicate high statistical significance for all correlations.

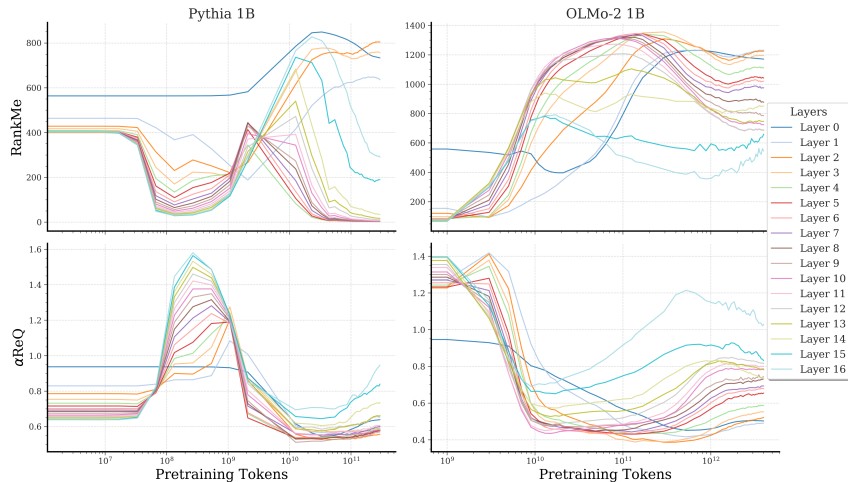

Figure 7: **Layerwise evolution mirrors the three phases.** Spectral metrics (`RankMe` and $\alpha$ReQ ) computed across intermediate layers during pretraining show that the three-phase pattern is consistent across network depth, justifying the use of last-layer representations for tracking global geometric dynamics. See Appendix for additional robustness analyses across samples, sequence lengths, and datasets.

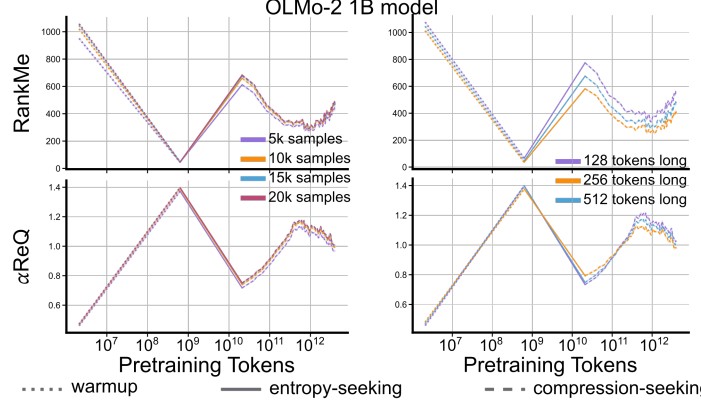

Figure 8: `RankMe` and $\alpha$ReQ computed for intermediate checkpoints of the OLMo-2-1B model using (**Left**) different number of samples, and (**Right**) sequence length. Shaded error bars indicate standard error about mean.

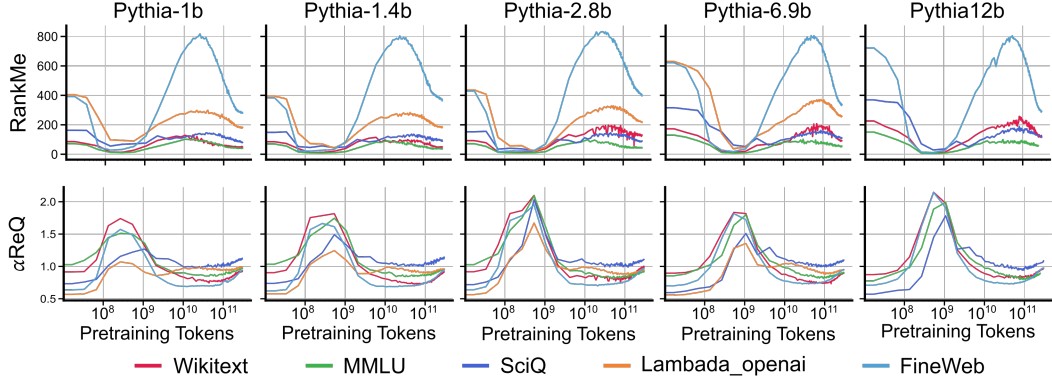

Figure 9: `RankMe` and $\alpha$ReQ computed for intermediate checkpoints of models from the Pythia family on different datasets.

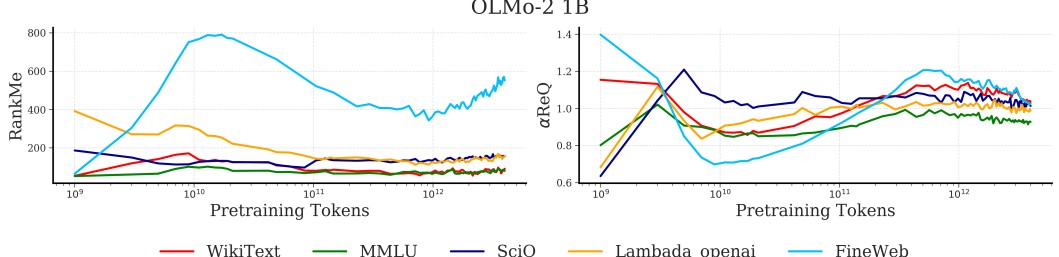

Figure 10: **OLMo spectral metrics are robust across datasets.** `RankMe` and $\alpha$ReQ computed for intermediate checkpoints of OLMo-2 1B model on different datasets, showing consistent phase patterns across evaluation data.

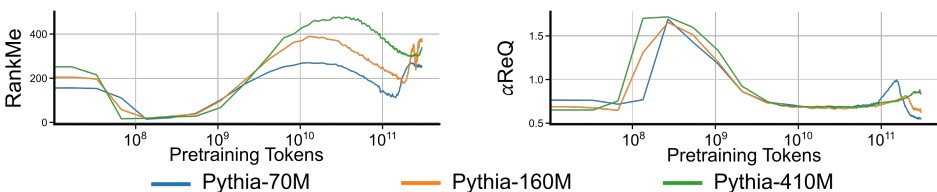

Figure 11: `RankMe` and $\alpha$ReQ computed for intermediate checkpoints of smaller models ($< 1$B) from the Pythia family on the FineWeb dataset.

## C.4 Control experiments verifying the necessity condition for multiphase learning dynamics

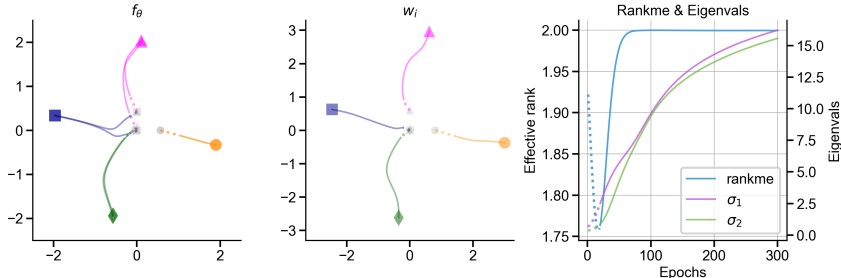

(a) Feature and weight dynamics in analytically-tractable model with uniform class distribution, i.e. each class has equal number of samples. Here, each class has 2 samples each.

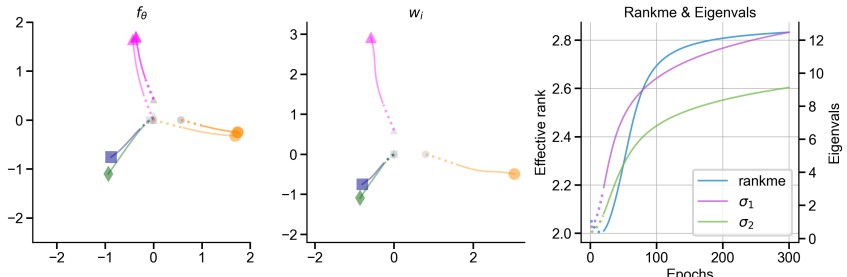

(b) Feature and weight dynamics in analytically-tractable model with no information bottleneck, i.e. feature dimensionality, $d$, is comparable to number of classes, $|\mathcal{V}|$. Here, $d = 3$ and $|\mathcal{V}| = 4$. Note that we only plot the first two dimensions for ease and consistency of visualization.

Figure 12: Negative control experiments analogous to Figure 4. Removing either the skewed class distribution or the information bottleneck gets rid of the three distinct phases of learning. In each case, the resulting dynamics is an initial "warmup" , followed by an "entropy-seeking" phase wherein effective rank continues to grow monotonically.

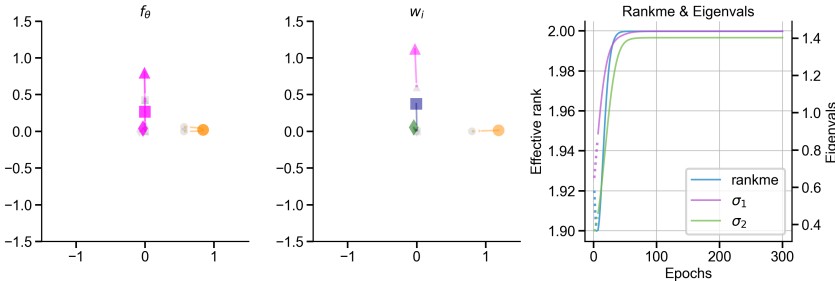

(a) Feature and weight dynamics in analytically-tractable model trained using MSE loss on a uniform class distribution, i.e. each class has equal number of samples. Here, each class has 2 samples each.

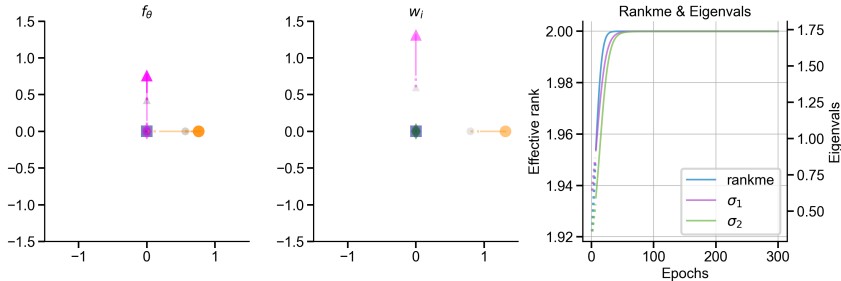

(b) Feature and weight dynamics in analytically-tractable model trained using MSE loss on a skewed class distribution. Note that only information about the most frequently occurring classes are learned.

Figure 13: Negative control experiments analogous to Figure 4, with mean squared error instead of cross-entropy as the training loss. In both uniform and skewed label distribution settings, the resulting dynamics is an initial "warmup" , followed by an "entropy-seeking" phase wherein effective rank grows monotonically and quickly saturates.

## C.5 Supervised finetuning

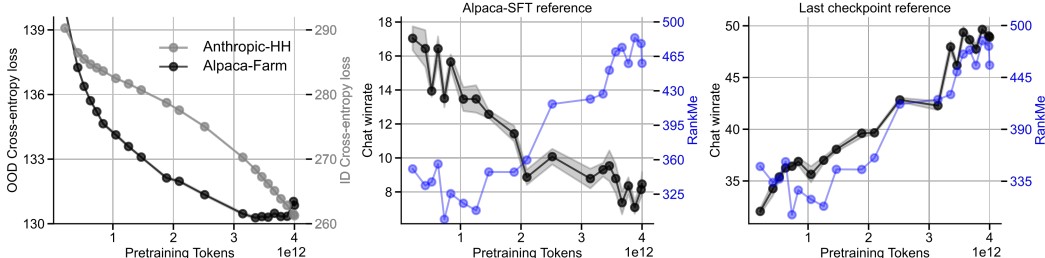

Figure 14: Loss and chat win rates after SFT on Anthropic-HH dataset. **(Left)** Cross-entropy loss on in-distribution (Anthropic-HH) test set and out-of-distribution (Alpaca-Farm-human-ANN chat) dataset. While in-distribution loss after SFT decreases monotonically, ood loss after SFT saturates or gets slightly worse with longer pretraining. **(Center)** Length-controlled chat win rates for Anthropic-SFT vs Alpaca-SFT version of a base model on AlpacaEval. Longer pretraining increases the sensitivity of the model's behavior to the SFT dataset. **(Right)** Length-controlled win rates for Anthropic-SFT version of intermediate base models compared to the Anthropic-SFT version of the final base model checkpoint. Models obtained from later in pretraining are equivalent chat models, demonstrating nearly 50% win rate compared to the final checkpoint. Choosing the ideal checkpoint to use for SFT requires navigating the tradeoff between an improvement in base model's capability (note an increased `RankMe`) and reduction in robustness with longer pretraining. Shaded bars indicate standard deviation computed over 5 seeds.

