# OpenReview forum: "Tracing the Representation Geometry of Language Models from Pretraining to Post-training"
_NeurIPS.cc/2025/Conference — NeurIPS 2025 poster_

### Official Review · Reviewer_YXFK · 2025-06-30

**Clarity:** 3
**Significance:** 3
**Originality:** 3
**Rating:** 4
**Confidence:** 3

**Summary:**

**Summary**
This paper introduces a **spectral analysis framework** to trace the evolution of representation geometry in LLMs across pretraining and post-training stages. Key contributions include:
1. **Non-monotonic pretraining phases**: Identifies three distinct geometric phases—*warmup* (rapid compression), *entropy-seeking* (manifold expansion correlating with n-gram memorization), and *compression-seeking* (anisotropic consolidation enabling generalization).
2. **Post-training dynamics**: Links SFT/DPO to *entropy-seeking* (improves in-distribution performance but reduces robustness) and RLVR to *compression-seeking* (enhances reward alignment but limits exploration).
3. Uses spectral metrics (RankMe, αReQ) to quantify representation geometry and connects findings to optimization dynamics and data skew.

**Questions:**

1. Would perturbing specific eigenvectors during training make the model reach flat minima, which is good for out-of-distribution generalization?
2. Would perturbing specific eigenvectors during training alter phase transitions or downstream performance?

**Ethical Concerns:**

["NO or VERY MINOR ethics concerns only"]

**Final Justification:**

The responses did not fully address the specific aspects I was curious about — which may indeed fall outside the main scope of the paper. I also acknowledge that the use of small-scale and limited instances of RLVR offer limited support for the claim that “RLVR **often** exhibits a ‘compression-seeking’ dynamic.” I will maintain my current positive score.

**Limitations:**

- **Dataset/model constraints**: Findings based on OLMo/Pythia families; generalization to other architectures and data remain untested.
- RankMe/αReQ may not capture fine-grained geometric shifts in early layers or task-specific subspaces.

**Quality:**

3

**Strengths And Weaknesses:**

**Strengths**
- **Novel framework**: Characterize LLM training via spectral geometry, revealing non-obvious phase transitions masked by loss curves.
- **Rigorous validation**: Demonstrates consistent phases across model families (OLMo, Pythia) and scales.
- **Practical insights**: Connects geometry shifts to capabilities (e.g., entropy-seeking → memorization; compression-seeking → generalization).
- **Methodological clarity**: Well-defined metrics (RankMe, αReQ) and ablation studies (e.g., Figure 4 validates phase drivers).

**Weaknesses**
- **Correlation vs. causation**: Observed geometry-behavior links are associative; causal mechanisms require further study.
- **Narrow task scope**: Experiments limited to n-gram memorization and simple QA; complex reasoning tasks unexplored.
- **Incomplete phase explanation**: Theoretical toy models (Section 3.3) simplify real LLM dynamics.

---

> ### Author Rebuttal · Authors · 2025-07-31
>
> We thank the reviewer for their encouraging comments and suggestions, and we are glad that they found our results to be novel, rigorous and clearly presented. Below, we address all the concerns and questions raised by the reviewer.
>
> ### Results on complex reasoning tasks
> We thank the reviewer for this suggestion. We have now conducted downstream task evaluations for checkpoints from intermediate pretraining stages on Lambada_openai, MMLU, and HellaSwag, and will add them in the appendix of our final paper. It is important to note that models solely undergoing pretraining often struggle with complex reasoning tasks, requiring specific post-training methods to achieve high performance. Therefore, their performance on such tasks is not a reliable indicator of their underlying capabilities. Consequently, Section 3.4, which presents results from post-training stages, focuses on evaluating model performance on complex tasks like chat (AlpacaEval) for SFT models and mathematical reasoning (AMC-23) for RLVR models.
>
> ### Utility of RankMe and $\alpha$-ReQ for intermediate layer representations
> While we focus on and present the results for the final layer representations, the spectral metrics that are used to quantify the representation geometry are generic and do not assume anything about the origin of the representations. Therefore, they can be readily applied to study changes in representation geometry in intermediate layers. To further illustrate this point, we performed experiments on the intermediate layer representations and observed similar trends. We will add these results in order to clarify the utility of the proposed metrics in the Appendix in the final version of our paper.
>
> ### Perturbing eigenvectors for better out-of-distribution generalization
> The relationship between better out-of-distribution generalization and flat minima implies a particular conditioning of the Hessian of the loss with respect to the model parameters. While the connection between this Hessian matrix and the eigenspectrum of the final layer representations is not immediately obvious, it becomes clear in cases where only the final readout weight matrix is adjusted for a new task. In such cases, perturbing specific eigenvectors can improve the optimization problem and, consequently, out-of-distribution generalization. However, more theoretical and empirical work is needed to fully understand the relationship between the geometry of final layer representations and a model's out-of-distribution performance after pretraining. We hope our proposed metrics and observed trends can inspire future strategies and regularization techniques to enhance out-of-distribution generalization.
>
> ### Perturbing eigenvectors to alter downstream performance
> This is an interesting question, highlighting the implications of our findings for mechanistic interpretability. While a full investigation of the causal relationship between the eigenspectrum of representations and model performance is beyond the scope of this work, we believe that our analytical results point to potential causal interventions. Specifically, we show that high-frequency token information is learned early in training and is encoded in the top eigendirections. Consequently, removing these directions would have minimal impact on downstream question-answering task performance. Conversely, preserving the top eigendirections while removing the tail of the eigenspectrum would retain most high-frequency token information but discard fine-grained dataset details, thereby negatively affecting downstream performance.
>
> Given the time constraints of the NeurIPS review process, we will defer a more rigorous analysis of how downstream task performance connects to this information segregation between the top eigendirections and the eigenspectrum tail to future work. Instead, we will add these preliminary results in the Appendix. These results will demonstrate the information segregation and the differential roles of the top eigendirections and the eigenspectrum tail on specific question-answering tasks, illustrating potential causal links between the eigenspectrum and the model's downstream behavior.
>
> ---
> We thank the reviewer once again for their overall positive review and are grateful for their helpful suggestions. We believe that adding the results on more diverse tasks during different phases in the pretraining stage, trends in representation geometry changes in intermediate representations and preliminary experiments demonstrating the effect of ablating the eigenspectrum tail on downstream task performance in the appendix will strengthen our paper and increase its impact. We hope that the reviewer finds our responses to be helpful, and are happy to clarify any further questions during the discussion phase.

---

> > ### Comment · Reviewer_YXFK · 2025-08-01
> >
> > I would like to thank the authors for their responses.
> >
> > > 1
> >
> >
> > Experiments over broader RLVR methods is of interest. Typically, RLVR does not utilize discounted accumulated reward - they typically consider 0-1 outcome reward-based policy gradients, or RAFT. GRPO and DAPO in this case are also preferable. I believe the consideration of these could further strengthen the paper's broader impact.
> >
> > > 2
> >
> >
> > Would you like to show your conducted experimental results you claimed here?
> >
> >
> > > Layer-wise analysis proposed by Reviewer 7SLc
> >
> >
> > I'm also interested in the layer-wise analyses you proposed to leash in the appendix. Would you like to show some results and analyses here?

---

> > > ### Author Response · Authors · 2025-08-04
> > > **Response to Reviewer YXFK**
> > >
> > > We thank the reviewer for their response. As we cannot add figures to our NeurIPS rebuttal (owing to the rebuttal guidelines), we have included the data table for the proposed experiments below.
> > >
> > > ### Spectral metrics for intermediate layers
> > > Given the length constraints, we only present the table for RankMe values of Pythia-1b model representations, derived from intermediate transformer blocks (referred to as model layers) below. We will include the Figures for RankMe and $\alpha$-ReQ for all models in the Appendix.
> > >
> > > | Tokens | Layer 1 | Layer 2 | Layer 3 | Layer 4 | Layer 5 | Layer 6 | Layer 7 | Layer 8 | Layer 9 | Layer 10 | Layer 11 | Layer 12 | Layer 13 | Layer 14 | Layer 15 | Layer 16 |
> > > |:--:|:---:|:---:|:---:|:---:|:---:|:---:|:---:|:---:|:---:|:---:|:---:|:---:|:---:|:---:|:---:|:---:|
> > > | 0M | 463.5 | 428.2 | 416.9 | 409.2 | 404.2 | 403.5 | 400.6 | 399.5 | 399.6 | 400.7 | 400.0 | 401.3 | 402.4 | 404.1 | 405.0 | 404.5 |
> > > | 128M | 367.5 | 230.3 | 170.0 | 134.0 | 109.0 | 90.2 | 76.5 | 64.7 | 56.0 | 49.0 | 43.0 | 38.1 | 34.5 | 31.1 | 28.4 | 29.2 |
> > > | 256M | 390.6 | 277.4 | 223.2 | 186.0 | 154.0 | 124.5 | 103.0 | 85.2 | 72.5 | 62.1 | 53.6 | 46.6 | 41.0 | 35.8 | 31.7 | 32.7 |
> > > | 512M | 327.5 | 251.1 | 219.3 | 206.1 | 176.7 | 149.6 | 129.1 | 114.7 | 102.9 | 91.6 | 81.7 | 73.2 | 65.9 | 59.3 | 53.1 | 54.2 |
> > > | 1B | 252.8 | 220.8 | 202.6 | 214.7 | 219.2 | 211.0 | 197.1 | 182.7 | 167.1 | 151.8 | 139.0 | 129.4 | 123.3 | 121.4 | 117.0 | 125.5 |
> > > | 2B | 187.7 | 265.8 | 287.1 | 344.5 | 412.9 | 442.2 | 445.1 | 442.2 | 428.9 | 395.1 | 370.7 | 335.8 | 314.7 | 297.7 | 275.2 | 293.5 |
> > > | 22B | 520.1 | 725.1 | 771.5 | 21.3 | 27.4 | 36.7 | 52.4 | 72.8 | 88.1 | 129.8 | 164.6 | 239.3 | 297.1 | 432.3 | 712.4 | 827.2 |
> > > | 42B | 587.0 | 757.3 | 776.0 | 5.6 | 6.8 | 8.4 | 11.1 | 14.8 | 17.6 | 27.0 | 36.7 | 63.6 | 89.7 | 145.0 | 440.7 | 765.0 |
> > > | 62B | 603.7 | 753.5 | 752.7 | 5.2 | 6.2 | 7.6 | 9.7 | 12.6 | 14.9 | 22.4 | 30.7 | 54.2 | 79.3 | 129.8 | 392.4 | 654.4 |
> > > | 82B | 622.1 | 756.4 | 740.9 | 4.4 | 5.2 | 6.1 | 7.6 | 9.5 | 11.0 | 16.0 | 21.7 | 38.8 | 58.4 | 97.3 | 308.7 | 578.9 |
> > > | 102B | 624.6 | 749.5 | 729.0 | 3.4 | 4.0 | 4.6 | 5.5 | 6.7 | 7.6 | 10.5 | 14.0 | 24.8 | 38.4 | 66.8 | 232.7 | 494.8 |
> > > | 122B | 635.0 | 757.1 | 731.4 | 3.3 | 3.8 | 4.3 | 5.1 | 6.1 | 6.8 | 9.4 | 12.3 | 21.9 | 34.5 | 61.7 | 220.7 | 442.2 |
> > > | 162B | 643.3 | 779.9 | 746.9 | 3.0 | 3.4 | 3.8 | 4.3 | 5.0 | 5.5 | 7.3 | 9.4 | 16.6 | 26.7 | 50.4 | 197.4 | 357.4 |
> > > | 202B | 646.7 | 795.2 | 756.1 | 2.7 | 3.0 | 3.3 | 3.7 | 4.2 | 4.6 | 5.9 | 7.5 | 13.2 | 21.5 | 42.4 | 188.6 | 309.1 |
> > > | 242B | 645.5 | 805.0 | 759.9 | 2.5 | 2.8 | 3.0 | 3.3 | 3.7 | 4.0 | 5.0 | 6.2 | 11.0 | 18.0 | 36.8 | 184.6 | 298.1 |
> > > | 282B | 635.4 | 803.8 | 756.2 | 2.4 | 2.6 | 2.9 | 3.1 | 3.5 | 3.7 | 4.7 | 5.8 | 10.1 | 16.6 | 34.4 | 189.0 | 289.9 |
> > >
> > > Note that the evolution of representation geometry is correlated across several layers (layer 4-16), with the three phases roughly overlapping in these layers. However, changes are more subtle in the very early layers. We believe that further work is essential to understand the mechanisms underlying this relationship of representation geometry across layers and how it affects model behavior.
> > >
> > >
> > > ### Perturbing eigenvectors to alter downstream performance
> > >
> > > In this experiment, we either remove or retain only the information contained in the subspace spanned by the top-k eigenvectors within the final hidden layer representations (before the LM head). Our results support the claim that preserving the top eigenvectors and removing the eigenspectrum tail significantly impacts model performance, whereas removing the top eigenvectors and preserving the tail has a negligible impact. We present evidence of this phenomenon for four different sizes of Pythia models on SciQ, and we will include a more detailed table in the Appendix.
> > >
> > > | Model   | Original | Top-10 removed | Top-10 only retained | Top-50 removed | Top-50 only retained |
> > > |:---:|:---:|:------:|:---:|:---:|:--:|
> > > | pythia-1b   | 0.838 | 0.849  | 0.225 | 0.835  | 0.318 |
> > > | pythia-1.4b | 0.866 | 0.855  | 0.232 | 0.859  | 0.324 |
> > > | pythia-2.8b | 0.884 | 0.880  | 0.219 | 0.873  | 0.317 |
> > > | pythia-6.9b | 0.896 | 0.893  | 0.202 | 0.906  | 0.327 |
> > >
> > > ### Beyond RLVR
> > >
> > > We appreciate the reviewer's insightful comment and suggestion to include broader RL methods. Although our work doesn't focus on RL methods for LLMs, we agree that such results would strengthen our paper. We will consider adding these results in the future, subject to time and space limitations.
> > >
> > > ---
> > > We thank the reviewer once again for their suggestions, and hope that these new results will help address their remaining questions.

---

> > > > ### Comment · Reviewer_YXFK · 2025-08-08
> > > >
> > > > Thank you. The responses did not fully address the specific aspects I was curious about — which may indeed fall outside the main scope of the paper. I also acknowledge that the use of small-scale and limited instances of RLVR offer limited support for the claim that “RLVR **often** exhibits a ‘compression-seeking’ dynamic.” I will maintain my current positive score.

---

> > > > > ### Author Response · Authors · 2025-08-08
> > > > > **Response by authors**
> > > > >
> > > > > We would like to thank the reviewer again for their comments and continued engagement during the discussion phase. We strongly believe that incorporating the reviewer's suggestions by adding the new results presented above will strengthen our paper and improve its overall impact.
> > > > >
> > > > > > The responses did not fully address the specific aspects I was curious about
> > > > >
> > > > > We were curious which aspect from our previous response the reviewer is referring to here, and if there is any specific result that we can add to alleviate this concern. We propose to add these new results in the appendix and add appropriate references to these results in the main text. Knowing which results we can strengthen will greatly help us in improving our paper and making it more well-rounded.
> > > > >
> > > > > > [...] the use of small-scale and limited instances of RLVR offer limited support for the claim that “RLVR often exhibits a ‘compression-seeking’ dynamic.”
> > > > >
> > > > > The reviewer is correct to point this out. We will update our claim to clearly indicate its scope, for example we will update this statement to be "In the models we studied in this work, RLVR often exhibits a 'compression-seeking' dynamic."
> > > > >
> > > > > We would like to thank the reviewer again for their constructive feedback through the review process and an overall positive rating of our work.

---

### Official Review · Reviewer_nZGd · 2025-07-03

**Clarity:** 3
**Significance:** 3
**Originality:** 2
**Rating:** 4
**Confidence:** 4

**Summary:**

The work examines the geometrical changes that occur during the training of state-of-the-art language models and their relationship to the capabilities acquired during training. It studies both pre-training checkpoints of OLMo and Pythia, as well as post-training checkpoints of the Tulu3 suite. It utilizes matrix entropy to characterize the geometry, specifically focusing on the last token representation of the model's final layer.  It shows that, in the course of pretraining, an expansion phase where the model memorizes n-gram statistics from the training corpus is followed by a compression phase, during which the model improves its performance in tasks like TriviaQA. SFT corresponds to an expansion phase where patterns from specific instructions are memorized at the expense of OOD generalization, and a compression in RLVR is attributed to the constraints imposed by RLVR on the exploratory capabilities of base models.

**Questions:**

Questions:

The authors omit some experimental details on how they preprocess the data to measure the entropy. For instance, the context length used in their experiments is not reported, nor is the impact of the sentence length on the measured entropy.
What is the impact of the sentence length on the measured entropy?

In Sec. 3.3 the authors should describe the dataset they use in more detail: is it text? Is the task language modeling? What are the classes they are talking about? What are their frequencies?

**Ethical Concerns:**

["NO or VERY MINOR ethics concerns only"]

**Final Justification:**

The authors addressed my concerns positively during the rebuttal phase (see the discussion with them below).

I consider the value of this contribution sufficient for acceptance to the NeurIPS conference.

**Limitations:**

Yes

**Paper Formatting Concerns:**

I do not find noticeable formatting issues

**Quality:**

3

**Strengths And Weaknesses:**

Strenghts

The work examines state-of-the-art models, offering an insightful analysis of the dynamics of the internal geometry of the representations, and sheds light on the connection between changes in the internal geometry of the last layer and the capabilities acquired during training.
The work is well written and easy to follow.

Weaknesses

By design, the work focuses on the last token at the last layer. This is fine, but the authors should clearly state, also in the discussion section and as a potential limitation of their work, that the compression-seeking and entropy-seeking phases are specific to the last-layer representations and not a general property of LLMs as a whole.
In the literature [1-3], there is evidence that the behavior of the geometry of the hidden representation is heavily layer-dependent. When the representation of the last layer expands, other internal representations may contract (and vice versa).
A discussion of this point should be added.

The plots are sometimes difficult to read. For instance, Fig.3 contains too many panels; three model sizes are enough, and more importantly, the colors of the profiles do not help the readability: with a double y-axis, colors should be used to distinguish the different quantities, such as SciQ and TriviaQA in the second line.

[1] Valeriani et al. The geometry of hidden representations of large transformer models.
[2] Ansuini et al. Intrinsic dimension of data representations in deep neural networks.
[3] Cheng et al. Emergence of a High-Dimensional Abstraction Phase in Language Transformers.

---

> ### Author Rebuttal · Authors · 2025-07-31
>
> We thank the reviewer for their helpful comments and suggestions, and we are glad that they found our paper to be well written and the results to be insightful. Below, we address all the concerns and questions raised by the reviewer.
>
> ###  Intermediate layer representations
> We agree with the reviewer that our work primarily focuses on the last token of the last layer, and thus, the identified phases are defined specifically based on the representation geometry changes in this representation space. The intermediate layer representations, as pointed out by the reviewer, do not necessarily exhibit the same trends. However, our analysis reveals similar trends in several intermediate layer representations, with the temporal evolution of last layer representations often serving as a leading indicator. This means intermediate layers undergo similar phases, but later in the pretraining stage. We will add these results into the Appendix and clarify the scope of our results in the Discussions section of the final paper.
>
> We defined our focus to be such because, as mentioned in Section 2.1, the last token in the last layer is critical for autoregressive models as: (i) it directly parameterizes the predictive distribution for subsequent tokens, (ii) it synthesizes information from the entire input context, reflecting the model’s contextual understanding, and (iii) it is frequently used as input for task-specific layers in downstream applications. This pivotal role allows us to directly relate spectral metrics to model behavior. But, that is not to say that the intermediate layers are unimportant, and for this reason, we will include those additional results in the updated Appendix.
>
> ### Spectral measure computation details
> We apologize for the lack of clarity in the methodology. We used 15k sequences from the Fineweb dataset and extracted the last layer representations after processing sequences of 512 tokens. We will add these details in Section 2.1 in the final version of our paper. The impact of sequence length and dataset on the spectral metrics is presented in Fig 7 and Fig 8 respectively. We will add references to these figures in Section 2.1 as well. We thank the reviewer for pointing this out.
>
> ### Section 3.3 experiment details
> We appreciate the reviewer's suggestion to add more detail about the analytically tractable experiment, as it will improve the paper's readability. In the experimental setup for Fig 4, our goal is to validate our theoretical findings regarding the role of cross-entropy loss, so we abstract away the complexities of natural language modeling. We define $f_{\theta}(.)$ to be an embedding model that receives inputs $x^{(i)}$'s, which are orthogonal vectors. Each $x^{(i)}$ is randomly assigned to one of four classes (represented by triangle, circle, square, and diamond in Fig 4A). The frequency distribution is skewed: two $x^{(i)}$’s belong to the triangle class, two to the circle class, and one each to the square and diamond classes (as shown in Fig 4A). For the control experiment in Fig 10(a), we use a uniform class distribution, with two samples per class.
>
> ### Fig 3 plots
> We apologize for the lack of clarity in the plots. We will reduce the number of panels in Fig 3 and make the necessary changes as suggested by the reviewer.
>
> ---
>
> We thank the reviewer once again for their comments and are grateful for their helpful suggestions. We believe that adding the results on representation geometry changes in intermediate representations in the appendix and clarifying the scope of our results in the discussions will strengthen our paper. We hope that the reviewer finds our responses to be helpful, and are happy to clarify any further questions during the discussion phase.

---

> > ### Comment · Reviewer_nZGd · 2025-08-02
> >
> > I thank the authors for their response, which has clarified some missing experimental details of their submission.
> >
> > I would also ask the authors to show in the appendix the dynamics of the matrix entropy of the hidden representations, comparing it to that of the last layer, and put a reference in the main text. **It is not generally true that the evolution of the last layer is a good indicator of the changes in the model internals.** For instance, see Fig. 3 in Valeriani et al. [1] or Fig. 2-right panel in Cheng et al. [2]. In these cases, phases of expansion of the last layer correspond to stages of compression of a large part of the hidden representations, and vice versa.\
> > In [1] and [2], the authors describe the geometry with the intrinsic dimension, which should be related to the matrix entropy. If the trend of the matrix entropy behaves differently, this should be explicitly stated, and evidence provided.
> >
> > That being said, since my concerns have been positively addressed, I raised my score to 4.
> >
> > [1] Valeriani et al., The geometry of hidden representations of large transformer models.
> >
> > [2] Cheng et al., Emergence of a High-Dimensional Abstraction Phase in Language Transformers.

---

> > > ### Author Response · Authors · 2025-08-04
> > > **Response to Reviewer nZGd**
> > >
> > > We thank the reviewer for their response. As we cannot add figures to our NeurIPS rebuttal (owing to the rebuttal guidelines), we have included a data table indicating the evolution of representation geometry in intermediate layer representations (derived from intermediate transformer blocks) of the Pythia-1B model.
> > > Given the space constraints, we only present the table for RankMe values of one model, but we will include the Figures for RankMe and $\alpha$-ReQ for all models in the Appendix.
> > >
> > > | Tokens | Layer 1 | Layer 2 | Layer 3 | Layer 4 | Layer 5 | Layer 6 | Layer 7 | Layer 8 | Layer 9 | Layer 10 | Layer 11 | Layer 12 | Layer 13 | Layer 14 | Layer 15 | Layer 16 |
> > > |:--:|:---:|:---:|:---:|:---:|:---:|:---:|:---:|:---:|:---:|:---:|:---:|:---:|:---:|:---:|:---:|:---:|
> > > | 0M | 463.5 | 428.2 | 416.9 | 409.2 | 404.2 | 403.5 | 400.6 | 399.5 | 399.6 | 400.7 | 400.0 | 401.3 | 402.4 | 404.1 | 405.0 | 404.5 |
> > > | 128M | 367.5 | 230.3 | 170.0 | 134.0 | 109.0 | 90.2 | 76.5 | 64.7 | 56.0 | 49.0 | 43.0 | 38.1 | 34.5 | 31.1 | 28.4 | 29.2 |
> > > | 256M | 390.6 | 277.4 | 223.2 | 186.0 | 154.0 | 124.5 | 103.0 | 85.2 | 72.5 | 62.1 | 53.6 | 46.6 | 41.0 | 35.8 | 31.7 | 32.7 |
> > > | 512M | 327.5 | 251.1 | 219.3 | 206.1 | 176.7 | 149.6 | 129.1 | 114.7 | 102.9 | 91.6 | 81.7 | 73.2 | 65.9 | 59.3 | 53.1 | 54.2 |
> > > | 1B | 252.8 | 220.8 | 202.6 | 214.7 | 219.2 | 211.0 | 197.1 | 182.7 | 167.1 | 151.8 | 139.0 | 129.4 | 123.3 | 121.4 | 117.0 | 125.5 |
> > > | 2B | 187.7 | 265.8 | 287.1 | 344.5 | 412.9 | 442.2 | 445.1 | 442.2 | 428.9 | 395.1 | 370.7 | 335.8 | 314.7 | 297.7 | 275.2 | 293.5 |
> > > | 22B | 520.1 | 725.1 | 771.5 | 21.3 | 27.4 | 36.7 | 52.4 | 72.8 | 88.1 | 129.8 | 164.6 | 239.3 | 297.1 | 432.3 | 712.4 | 827.2 |
> > > | 42B | 587.0 | 757.3 | 776.0 | 5.6 | 6.8 | 8.4 | 11.1 | 14.8 | 17.6 | 27.0 | 36.7 | 63.6 | 89.7 | 145.0 | 440.7 | 765.0 |
> > > | 62B | 603.7 | 753.5 | 752.7 | 5.2 | 6.2 | 7.6 | 9.7 | 12.6 | 14.9 | 22.4 | 30.7 | 54.2 | 79.3 | 129.8 | 392.4 | 654.4 |
> > > | 82B | 622.1 | 756.4 | 740.9 | 4.4 | 5.2 | 6.1 | 7.6 | 9.5 | 11.0 | 16.0 | 21.7 | 38.8 | 58.4 | 97.3 | 308.7 | 578.9 |
> > > | 102B | 624.6 | 749.5 | 729.0 | 3.4 | 4.0 | 4.6 | 5.5 | 6.7 | 7.6 | 10.5 | 14.0 | 24.8 | 38.4 | 66.8 | 232.7 | 494.8 |
> > > | 122B | 635.0 | 757.1 | 731.4 | 3.3 | 3.8 | 4.3 | 5.1 | 6.1 | 6.8 | 9.4 | 12.3 | 21.9 | 34.5 | 61.7 | 220.7 | 442.2 |
> > > | 162B | 643.3 | 779.9 | 746.9 | 3.0 | 3.4 | 3.8 | 4.3 | 5.0 | 5.5 | 7.3 | 9.4 | 16.6 | 26.7 | 50.4 | 197.4 | 357.4 |
> > > | 202B | 646.7 | 795.2 | 756.1 | 2.7 | 3.0 | 3.3 | 3.7 | 4.2 | 4.6 | 5.9 | 7.5 | 13.2 | 21.5 | 42.4 | 188.6 | 309.1 |
> > > | 242B | 645.5 | 805.0 | 759.9 | 2.5 | 2.8 | 3.0 | 3.3 | 3.7 | 4.0 | 5.0 | 6.2 | 11.0 | 18.0 | 36.8 | 184.6 | 298.1 |
> > > | 282B | 635.4 | 803.8 | 756.2 | 2.4 | 2.6 | 2.9 | 3.1 | 3.5 | 3.7 | 4.7 | 5.8 | 10.1 | 16.6 | 34.4 | 189.0 | 289.9 |
> > >
> > >
> > > We agree with the reviewer that the last layer alone may not fully indicate changes in intermediate representations. However, our results show a correlation in the evolution of representation geometry across several layers (4-16), with the three observed phases largely overlapping in these layers. This suggests that the evolution of the last layer representations can indeed be indicative of changes in these deeper layers.
> > > As the reviewer noted, changes in very early layers are more subtle and may be temporally lagged versions of later layers. Consequently, the "smoothness-seeking" phase observed in later layers might correspond to an "entropy-seeking" phase in the early layers. We believe further research is crucial to understand the mechanisms underlying this cross-layer relationship in representation geometry and its impact on model behavior.
> > >
> > > Following the reviewer’s suggestion, we will add these results to the Appendix and refer to them in the main text. We will also cite previous related works by Valeriani et al. and Cheng et al., and discuss the relationship between our results and theirs in the Appendix. We thank the reviewer once again for bringing these works to our attention.
> > >
> > > ---
> > > We are glad that our rebuttal positively addressed the reviewer’s concerns and hope that these new results will resolve any remaining questions.

---

> > > > ### Comment · Reviewer_nZGd · 2025-08-06
> > > >
> > > > Thank you for the answer. As I said above, I considered my concerns addressed. For this reason, I (already) decided to increase the score.

---

### Official Review · Reviewer_7SLc · 2025-07-06

**Clarity:** 4
**Significance:** 3
**Originality:** 3
**Rating:** 5
**Confidence:** 4

**Summary:**

This paper investigates the evolution of representation geometry in LLMs throughout pretraining and post-training using spectral analysis methods. The authors employ effective rank (RankMe) and eigenspectrum decay rate (αReQ) to characterize representation complexity and identify three distinct geometric phases during pretraining: "warmup," "entropy-seeking," and "compression-seeking." They demonstrate correlations between these phases and model capabilities, with entropy-seeking associated with memorization and compression-seeking with generalization. The analysis extends to post-training methods (SFT, DPO, RLVR), showing that different alignment techniques induce distinct geometric transformations. The work provides both empirical validation across multiple model families (OLMo, Pythia, Tülu-3.1) and theoretical grounding through analytically tractable toy models.

**Questions:**

1. **Mechanistic Validation**: Can the authors provide more direct evidence that the geometric phases causally influence capability acquisition? For instance, interventions that artificially modify representation geometry during training could test whether this affects the emergence of memorization or generalization capabilities.

2. **Robustness Analysis**: How sensitive are the identified phases to hyperparameter choices (learning rate schedules, batch sizes, model architectures)? Understanding this robustness is crucial for determining the generality of the findings.

3. **Alternative Metrics**: Have the authors considered other measures of representation geometry (e.g., intrinsic dimensionality, alignment metrics, or information-theoretic measures)? Validation with alternative metrics would strengthen confidence in the conclusions.

4. **Layer-wise Analysis**: The paper focuses on last-layer representations, but how do these geometric phases manifest across different layers? Understanding the spatial distribution of these dynamics could provide deeper mechanistic insights. For example, as mentioned before, the recent work "Layer by Layer: Uncovering Hidden Representations in Language Models" (arxiv:2502.02013) also uses information-theoretic and geometric measures (including effective rank) to analyze LLM representations, but focuses on spatial analysis across layers rather than temporal evolution during training. Could the authors discuss the similarities and differences between their temporal geometric phase framework and this spatial layer-wise analysis?

6. **Intervention Studies**: Can the geometric insights be used to design more effective training strategies? For example, could learning rate schedules be adapted based on geometric phase transitions to optimize the memorization-generalization trade-off? (similar to "Seq-VCR: Preventing Collapse in Intermediate Transformer Representations for Enhanced Reasoning" (arxiv:2411.02344) which applied matrix entropy to increase reasoning perforemce)

**Ethical Concerns:**

["NO or VERY MINOR ethics concerns only"]

**Final Justification:**

They addressed most of my concerns. I still have concerns about the evaluation of memorization and the limited theoretical contributions

**Limitations:**

The authors adequately address limitations in Section 5, acknowledging that their findings may be specific to the investigated model families and datasets. However, they could more explicitly discuss the limitations of their spectral analysis approach and the potential for other geometric measures to reveal different patterns. The correlational nature of the evidence and the need for stronger causal analysis should also be emphasized more prominently. Additionally, focusing on last-token representations may overlook important dynamics occurring in intermediate layers or attention mechanisms.

**Quality:**

3

**Strengths And Weaknesses:**

## Strengths

**Novel Analytical Framework**: The application of spectral geometry to understand LLM training dynamics represents a significant methodological contribution. Moving beyond loss curves to examine representation structure provides valuable insights into the learning process that are not captured by traditional optimization metrics.

**Comprehensive Empirical Validation**: The authors demonstrate consistency across multiple model families (OLMo, Pythia, Tülu-3.1) and scales (1B-13B parameters).

**Mechanistic Understanding**: The connection between geometric phases and specific capabilities is well-supported. The correlation between entropy-seeking phases and n-gram memorization, followed by compression-seeking phases and improved long-context reasoning, provides actionable insights for understanding when different capabilities emerge during training.

**Theoretical Grounding**: The toy model analysis in Section 3.3 effectively demonstrates how cross-entropy optimization under information bottleneck conditions can reproduce the observed multiphase dynamics. The identification of primacy bias and selection bias as key mechanisms provides a theoretical foundation for the empirical observations.


## Weaknesses

**Limited Theoretical Rigor**: While the toy models provide intuitive explanations, the formal connection between these simplified settings and actual LLM training remains tenuous. The paper would benefit from more rigorous theoretical analysis establishing when and why these phases must occur in realistic training scenarios.

**Narrow Evaluation of Memorization**: The distributional memorization metric based on n-gram alignment, while useful, captures only one aspect of memorization. The paper's conclusions about memorization vs. generalization would be stronger with additional memorization measures and more diverse downstream tasks.

**Causality vs. Correlation**: The paper establishes strong correlations between geometric phases and capabilities but does not demonstrate causality. It remains unclear whether geometric changes drive capability emergence or vice versa, limiting the actionability of the insights.

**Post-training Analysis Depth**: The post-training analysis, while interesting, lacks the depth and mechanistic understanding provided for pretraining. The connections between different alignment objectives and geometric transformations could be explored more thoroughly.

**Methodological Limitations**: The focus on last-token representations may miss important dynamics in other parts of the networks (For example, analyze the intermediate layers as in "Layer by Layer: Uncovering Hidden Representations in Language Models" (arxiv:2502.02013)"

---

> ### Author Rebuttal · Authors · 2025-07-31
>
> We thank the reviewer for their encouraging comments and helpful suggestions, and we are glad that they found our results to be novel, well-supported and theoretically grounded. Below, we address all the concerns and questions raised by the reviewer.
>
>
> ### Mechanistic Validation
>
> This is a great question, and forms an important next step to our current work. While we have deferred investigating causal relationships between eigenspectrum of representations and model behavior to future work, we believe that our analytical results shed light into potential causal interventions that can be designed. Specifically, we demonstrate that the high-frequency token information is learned earlier in training and mainly represented in the top eigendirections. Consequently, removing these directions would have minimal impact on downstream question-answering task performance. On the other hand, keeping the top eigendirections and removing the tail of eigenspectrum would imply preserving most of the information about high-frequency tokens and removing fine-grained information about the dataset, thereby hurting downstream performance. While the concepts of memorization and generalization are loosely connected to this information segregation between the top eigendirections and the tail of eigenspectrum, given the time constraints of the NeurIPS review process, we defer a more rigorous analysis to future work. Instead, we propose to add preliminary results demonstrating the information segregation and the differential role of the top eigendirections and the eigenspectrum tail on the task performance in the Appendix, to demonstrate the potential causal links between eigenspectrum and behavior. We thank the reviewer for this suggestion.
>
> ### Robustness analysis
>
> While we agree with the reviewer's suggestion that it would help to understand the impact of hyperparameter choices, our compute budget limitations prevented us from pretraining large-scale models ourselves. Instead, we used open-source models and checkpoints, demonstrating consistent non-monotonic spectral evolution across Pythia and OLMo suites with varying architectures and scales. Our results are limited to existing hyperparameter configurations of these model families. However, we believe these results encourage future work on understanding the impact of design choices on the multi-phase learning dynamic.
>
> For smaller-scale models, we have preliminary results where we trained GPT-2 models from scratch on Fineweb and observed the emergence of multi-phase learning dynamics across various "good" hyperparameter configurations (monotonic loss decrease, reasonable downstream performance). We will include an Appendix figure illustrating these results. Taken together, we believe that we can broadly say that these phases are present in well-performing LLMs trained with next-token prediction and cross-entropy loss, deferring further investigation into the impact of hyperparameters to future work.
>
>
> ### Alternate metrics
>
> While the reviewer is correct in pointing out the existence of various metrics for quantifying representation geometry, we focused on two that rely on linear decomposition. Our choice was motivated by the direct relationship between linear decomposition of the representation space and learning dynamics under gradient descent. This connection allows us to link our findings to empirically observed model behaviors and provide a well-grounded theoretical explanation.
>
> In contrast, metrics like intrinsic dimensionality identify a non-linear transformation of the representation space that explains most of its variance. While useful for understanding the underlying representation manifold, these metrics are not directly connected to gradient descent properties, making it harder to draw concrete conclusions about model behavior.
>
> Furthermore, recent work by Skean et al. [1] (the paper recommended by the reviewer) demonstrates theoretical connections and strong empirical correspondence between several information-theoretic, geometric and spectral metrics in LLMs. Therefore, we believe our results, combined with those of Skean et al., offer a more comprehensive view of representation geometry evolution in LLMs across pretraining and post-training stages. We will add this point in the discussion section of our paper, along with the citation to the recent work from Skean et al.
>
>
>
> ### Layer-wise analysis
>
> We once again thank the reviewer for highlighting the recent work from Skean et al. on layer-wise representation geometry. This interesting work offers a complementary perspective on model behavior, and we will cite it and discuss its connection to our own in the updated discussion section of our paper. Notably, our analysis shows similar trends in intermediate layer representations, with the last layer's temporal evolution often serving as a leading indicator. Specifically, most intermediate layers exhibit similar phases, but later in pretraining. We will add these results in the Appendix.
>
> While Skean et al. examined similarity across model architectures and modalities from an information processing viewpoint, our work concretely links representation geometry metrics to downstream model behavior and provides a foray into potential mechanisms underlying the representation geometry changes during pretraining and post-training stages. Thus, our findings offer a complementary view of how the training strategies lead to representational changes in LLMs and how those lead to emergent behaviors.
>
>
> ### Intervention studies
>
> We thank the reviewer for this insightful suggestion and pointing to this work by Arefin et al. We agree with the reviewer that our findings on multiple phases could lead to spectral interventions and regularization strategies for accelerating training and improving LLM generalization. We hypothesize that some of this regularization might be implicit in current objective functions, such as the equivalence between the DPO objective and contrastive loss that reveals an implicit entropy-seeking prior. The interplay between these implicit biases and explicit regularization warrants further study, which we believe is beyond the scope of this paper and leave for future work. However, we agree with the reviewer that mentioning this potential impact of our results will strengthen our paper, and we will add this point and a citation to Arefin et al. in the discussion section.
>
> ---
> We thank the reviewer once again for their overall positive rating and are grateful for their helpful suggestions. We believe that incorporating our results from preliminary experiments altering the eigenspectrum and GPT-2 training in the appendix as well as discussing the recent works from Skean et al. and Arefin et al. will strengthen our paper and improve its impact. We hope that the reviewer finds our responses to be helpful, and are happy to clarify any further questions during the discussion phase.
>
> **References:**
> 1. Skean, O., Arefin, M.R., Zhao, D., Patel, N.N., Naghiyev, J., LeCun, Y. and Shwartz-Ziv, R., 2025. Layer by Layer: Uncovering Hidden Representations in Language Models. In Forty-second International Conference on Machine Learning.

---

> > ### Comment · Reviewer_7SLc · 2025-08-07
> >
> > Thank you for your detailed response. They addressed most of my concerns.

---

> ### Author Response · Authors · 2025-08-09
> **Response to Reviewer 7SLc**
>
> We thank the reviewer for their response and for engaging with us during the discussion phase. We appreciate the positive feedback and believe that the revisions made in response to the reviewer’s suggestions will enhance the clarity and overall quality of the paper. We are grateful for the reviewer’s time and constructive comments.

---

### Official Review · Reviewer_61qF · 2025-07-06

**Clarity:** 2
**Significance:** 2
**Originality:** 3
**Rating:** 3
**Confidence:** 5

**Summary:**

This paper studies final token representations, in the last layer of an LLM over the timecourse of pre and post training. By looking at summary measures related to the rank of the covariance of these final token representations the work presented identifies three phases of training - warmup, entropy seeking, and compression seeking. This leads to claims regarding what is acquired in each phase of training, and how this relates to model behaviour.

**Questions:**

Are there any significant correlation results between the representational measures used here and performance on a task?

What is the relationship between the memorisation measure and dependency length? What is the difference between long-distance dependencies and long-context factual knowledge?

**Ethical Concerns:**

["NO or VERY MINOR ethics concerns only"]

**Final Justification:**

In the general case, I appreciate the willingness of the authors to engage with the review process. However at this time I still have concerns about the nature of the statistical testing performed - which I believe is key to making the claims this paper hopes to. I also still have concerns about whether the experiments relating to context length are sufficiently well controlled.

Overall my review noted that this paper is quite dense, and the rebuttal has left that largely unaddressed. While this work is interesting it currently introduces a lot of ideas for 9 pages, as a result information that seems critical to the reader has been omitted in places (eg. that the infini-gram model is infact a quad gram, or that previous work has suggested SciQ is stable under sliding window attention).

Having looked at the other reviews I believe my original score stands - I do not think this work is currently ready for publication, but look forward to a future more polished version of it.

**Limitations:**

As mentioned above in strengths/weaknesses, the limitations section discusses the correlational nature of the evidence here. However the results themselves include virtually no statistical testing, and so appear more associative & speculative, than correlational.

It might be worth noting in the limitations section that results seem to focus on representations in only the final token, in only the final layer. While this allows the authors to draw some interesting conclusions it is distinct from studying the model in toto.

**Quality:**

3

**Strengths And Weaknesses:**

The work presented here is interesting, and seeks to address a number of important questions in interpretability. The use of a geometric framing presents some novelty -- the methods used do not appear novel to this work, but the experiments here apply them to a broader set of questions and models. In the general case the geometric perspective is distinct from mechanistic interpretability and somewhat underrepresented in the literature. However, the paper itself is a bit dense - trying to fit an large number of results into 9 pages sometimes at the expense of clarity. Additionally many of the claims feel somewhat out of scope with the results on which they're based. Below I highlight a couple of high-level concerns before providing some more detailed feedback on the results.

In the limitations section the authors note "our findings are primarily correlational in nature; additional theoretical and empirical investigation is required to develop deeper mechanistic insights and establish causal connections between these observed geometric dynamics and model behavior." However it appears that the findings throughout this paper use virtually no statistical testing. The only correlational results I noticed were related to the memorisation measure which is based around a spearman correlation. Given that the Pythia suite releases multiple initialisations for each size the authors could have tried to correlate representation complexity for different seeds of a model with performance on a benchmark. This would constitute more compelling evidence of the link between their representational measures and performance. Throughout there are a number of claims that appeal to the time-series plots but could be made stronger with statistical testing; to give one example line 270 claims "This decline in pass@256 performance as training progresses, reinforces prior work ... suggesting that RLVR constraints the exploratory behavior of ... base models while amplifying some pre-existing behaviors of the base model" -- there doesn't appear to be a quantification of the relationship between pass@256 performance and representation structure. I broadly agree with the authors that correlational results can be used to preform interesting explorational analyses, but currently this work seems to present results that are more speculative & associative than correlational.

While the high-level methods are well-explained I lack some clarity of the particular instantiation for the experiments presented here. How many samples are used to compute the estimate, and with respect to what dataset? There appear to be some benchmarking experiments for different sample sizes in the appendix, but I'm unsure if a single reference corpus is used for computing RankMe estimates or if, for example, in section 3.4 estimates are computed with respect to AH and AF respectively, or with resepect to a combination of the two. This would seem to affect the interpretation of the results in terms of whether models expand or compress in the general case or only w.r.t. the data on which they are fine-tuned.

Some claims seem weakly related to the experiments performed. As an example line 304: "This geometric refinement process is consistent with and may offer an explanation for phenomena like grokking." The results related to grokking appear to be based on results from a multiple choice dataset (triviaQA) - it seems likely the sudden jump in performance is more related to the fact that it is multiple choice than grokking effects like those studied on models of modular addition. Given the large number of results already in the paper, authors could remove more speculative lines of argumentation like this to improve clarity. The associative nature of some claims is reaffirmed by the inclusion of plots that have "representational complexity" on the y axis, and pre/post training on the x axis neither of which have units. The version of this plot in figure one also includes what appear to be suggestions about what happens in each phase -- I'm not quite sure what the evidence for some of these names are (authors draw a distinction between when "long-range dependencies" are learned, and when "long-context factual knowledge" is learned maybe based on a minor increase in "representational complexity" or increased performance on the aforementioned QA task). This visual also sometimes includes question marks to make clear the claims are speculative (e.g. "preference memorisation?").

In general this is interesting work, that would benefit from a more-focussed argument with more tightly scoped claims. Or which may be better suited to a venue that publishes longer format works to allow the authors the space necessary to fully unpack the breadth of ideas presented here.

_______________________
Detailed Feedback

Looking at the results in section 3.2; the ngram correlation is introduced as a measure of memorisation, but the analysis discusses the dependency length learned by the model - it is non-trivial to relate these two. A counter argument could be that learning local dependencies is reflected in correlation with a bigram or trigram lm, and longer dependencies correlate with larger n's approaching the infinite case -- see work like Someya et al. 2025 for an example of more nuanced treatment of dependency length. There are also substantial differences between the performance on two different factual knowledge benchmarks over the course of training - these benchmarks differ primarily in whether or not they are multiple choice. The substantial difference in empirical results between these two is attributed to the dependency length needed to answer the question, an assertion that is presupposed rather than being argued for, and which is still not clearly related to the "memorisation metric".

Section 3.3 claims to show: "Taken together, these results suggest that gradient-based cross-entropy optimization dynamics under specific training conditions may result in non-monotonic changes in representation geometry we observed in LLMs." It would seem one of the natural control conditions for this claim would be to use an objective other than cross-entropy, however I cannot find results to this effect in the main paper or supplementary materials. Short of this it is unclear the role that cross entropy plays or the interaction between the objective function and the two independent variables used here (non-uniformity, and an information bottleneck).

Section 3.4 looks at how post-training affects in-distribution and out-of-distribution performance on two post-training datasets. It is worth noting that while the AH and AF datasets are distinct, the claim that they are "out of distribution" from each other is not motivated. I'll note the evaluation procedure at line 250 is too obliquely explained for me to have a strong intuition for what the decrease in winrate means. Additionally the formal relation exemplified by equation 5 is currently introduced too briefly for me to have strong intuitions about its importance or relevance to the narrative through-line.

_______________________
References

Someya et al. 2025
https://arxiv.org/pdf/2506.05136

---

> ### Author Rebuttal · Authors · 2025-07-31
>
> We thank the reviewer for their astute observations, encouraging comments and helpful suggestions, and we are glad that they found our work to be interesting and timely. Below, we address all the concerns and questions raised by the reviewer.
>
> ### Statistical analysis
>
> We agree that adding statistical testing will strengthen our results. Below, we present spearman correlation analysis results for the following results:
>
> 1. **Correlation with task performance:** We compute the correlation between the spectral metrics and downstream task performance (SciQ and TriviaQA) across different pretraining checkpoints for Pythia models.
>
> SciQ
> | Model | $\alpha$-ReQ (p-value) | RankMe (p-value) |
> | :--- | :---: | :---: |
> | pythia-1b | 0.810 (1.50e-5) | -0.759 (1.04e-4) |
> | pythia-1.4b | 0.668 (1.29e-3) | -0.713 (4.18e-4) |
> | pythia-2.8b | 0.694 (6.88e-4) | -0.635 (2.63e-3) |
> | pythia-6.9b | 0.837 (4.20e-6) | -0.885 (2.19e-7) |
> | pythia-12b | 0.836 (4.42e-6) | -0.839 (3.79e-6) |
>
> TriviaQA
> | Model | $\alpha$-ReQ (p-value) | RankMe (p-value) |
> | :--- | :---: | :---: |
> | pythia-1b | 0.915 (8.263e-11) | -0.899 (9.844e-12) |
> | pythia-1.4b | 0.924 (9.251e-12) | -0.915 (2.361e-12) |
> | pythia-2.8b | 0.905 (9.586e-11) | -0.898 (3.645e-11) |
> | pythia-6.9b | 0.954 (8.969e-11) | -0.898 (4.675e-15) |
> | pythia-12b | 0.840 (2.223e-3) | -0.790 (6.428e-4) |
>
> 2. **Correlation with pass@256 performance:**  Since both RankMe and pass@256 values are nearly monotonically decreasing, the spearman rank correlation is ~1. We will extend our RLVR analysis to more checkpoints and compute the correlation for these results, and update Fig 5C.
>
> We propose to add these results to the Appendix, in order to strengthen our claims about the correlational behavior of our findings.
>
> ### Experiment details for computing spectral metrics
> We apologize for the lack of clarity in these experimental details. For all experiments, we used 15k sequences from the Fineweb dataset and extracted the last layer representations after processing sequences of 512 tokens. We will add these details in Section 2.1 and amend it to have references to the ablation experiments in Fig 7 and Fig 8. We hope that this also clarifies the related concerns about the results in Section 3.4.
>
> ### Clarification about grokking
> While we agree the mechanism of grokking here differs from those studied in models of modular addition, we would like to clarify that TriviaQA is a factual question-answering dataset, not multiple-choice. Consequently, TriviaQA performance is measured by Exact Match metric via LLM harness. Nevertheless, we concur that "grokking" is an overloaded term and used it to describe a sudden increase in 0-shot task performance. To improve clarity and avoid overstatement, we will replace "grokking" with "rapid increase in 0-shot performance" in the text.
>
> ### Clarification about Fig 1
> Fig 1C summarizes our key findings graphically without diving into the technical details, using "representation complexity" to denote representation space entropy and only distinguishing between pre-training and post-training stages on the x-axis. While we believe it offers a high-level overview, we acknowledge the reviewer's feedback regarding clarity and avoiding speculation. Therefore, we will clearly differentiate between established and speculative claims in the figure and note this distinction in the caption.
>
> ### Clarification about Section 3.2 results
> 1. **Infini-gram results:** We apologize for the lack of clarity in connecting our results to the claim about short-sequence memorization. The infini-gram model, despite its name, does not use a very large context window for predictions. It is a generalization of the n-gram model, where 'n' is not fixed and instead set to the length of the longest suffix found in the corpus. For samples from the TriviaQA dataset, we analyzed the suffix length from the question used to predict tokens in the answer and found that the mean was 4.21 tokens (median 4.0, max 12). A more detailed breakdown of the frequency of different suffix lengths is presented below.
> | Suffix Length | Frequency (%) |
> | :--- | :---: |
> | $\le$3 | 25.41 |
> | 4 | 46.61 |
> | 5 | 16.71 |
> | 6 | 6.08 |
> | 7 | 2.40 |
> | 8 | 1.46 |
> | $>$8 | 1.34 |
>
> This analysis indicates that the infini-gram model's predictions primarily depend on short- to mid-range dependencies. While we acknowledge the reviewer's valuable suggestion of a more nuanced analysis using bigrams and trigrams, the computational burden of training several n-grams on the entire pretraining dataset is not feasible within the given rebuttal timeframe. Therefore, we have restricted our analysis to using infini-grams to calculate the memorization metric, and we will clarify the precise meaning of this measure in the updated manuscript.
>
> 2. **Context length dependencies for SciQ and TriviaQA:** The reviewer is correct that a key difference between SciQ and TriviaQA is that SciQ is a multiple-choice task, whereas TriviaQA is not. However, TriviaQA is included in the LongBench dataset [1] and is considered to require long-context understanding. In contrast, Lou et al. showed that SciQ accuracy, unlike TriviaQA performance, remains relatively stable even when using sliding window attention (Tables 2 & 4 in [2]), indicating that long-context information is not essential. Based on these results, we believe the performance discrepancy between the two benchmarks is in part due to the different dependency lengths needed to answer questions in their respective datasets. But, there may also be an effect due to the increased reasoning requirement to answer questions in the TriviaQA dataset, and we will note this as another possible interpretation in the updated text.
>
> We hope that this clarification about the context-length required for the two tasks and the relationship between the memorization metric and short-context understanding ability will help resolve the logical jump pointed out by the reviewer.
>
> ### Control experiment for clarifying the role of cross-entropy
> We agree that using an objective other than cross-entropy, e.g. mean squared error (MSE), would be a natural control for our claim. We extended our theoretical analysis in Appendix B to derive the learning dynamics under an MSE loss. Simulations in the analytically tractable settings show that the eigenvalues monotonically increase (with $\sigma_1$ increasing faster than $\sigma_2$ initially) and eventually saturate. Consequently, the RankMe first decreases (warmup phase), then increases (entropy-seeking phase), and finally saturates. Notably, unlike the cross-entropy case, there is no smoothness-seeking phase when using MSE. This result further supports our claim that cross-entropy loss's learning dynamics are a key factor in the multiphase changes in representation geometry. We will add the MSE learning dynamics derivation and plots to the Appendix.
>
> ### Clarification about Section 3.4 results
> 1. **Ood nature of AH and AF:** As mentioned in Section A.2, AH is a chat dataset aimed at incorporating helpfulness and harmlessness, while AF is a generic chat dataset. The out-of-distribution (ood) nature spawns from the fact that the chat responses in the dataset are qualitatively different. While responses in AF are more geared towards short and concise responses, responses in AH are more conversational, clarifying and thought-provoking. Furthermore, the prompts in AF are relatively straightforward and atomic in nature, i.e. they ask the model to do a single task, whereas the prompts in AH are complex user queries and often require deeper reasoning and understanding of the underlying intent. These qualitative differences make the two datasets distinct and inherently ood to each other.
>
> 2. **Intuitive explanation of the chat winrate results:** The winrate reported in Fig 5B (bottom) is computed by comparing the AH fine-tuned and AF fine-tuned model responses for prompts from a third, novel dataset: AlpacaEval. This comparison is done with a larger LLM judge which picks the more appropriate answer between the two. It is worth noting that AlpacaEval, which has prompts of similar style and complexity to AF, is more out-of-domain for AH fine-tuned models than AF fine-tuned models. This is reflected in the low winrates (6-14%) of AH fine-tuned models with respect to AF fine-tuned models. Our key observation is the decrease in this winrate with more pretraining of the base model, suggesting that ``overtrained'' models (those with higher RankMe values) perform better on in-domain tasks but worse at generalizing to out-of-domain data.
>
> 3. **Intuitive explanation of the DPO objective reformulation:** Equation 5 establishes the equivalence between the DPO objective and the contrastive learning objective. This equivalence suggests similar learning dynamics for both, which could explain the empirically observed entropy-seeking phase during the DPO training stage. We aim to further elaborate on this relationship and connect DPO learning dynamics to previous works on self-supervised learning dynamics in the Appendix of our paper's final version.
>
> ---
> We thank the reviewer once again for their astute comments and are extremely grateful for their helpful suggestions. We believe that adding the statistical testing results and clarifications will strengthen our work and improve the clarity of the presentation. We hope that the reviewer finds our responses to be helpful, and are happy to clarify any further questions during the discussion phase.
>
> **References:**
> 1. Bai, Y., Lv, X., Zhang, J., Lyu, H., Tang, J., Huang, Z., Du, Z., Liu, X., Zeng, A., Hou, L. and Dong, Y., 2023. Longbench: A bilingual, multitask benchmark for long context understanding. arXiv preprint arXiv:2308.14508.
> 2. Lou, C., Jia, Z., Zheng, Z. and Tu, K., 2024. Sparser is faster and less is more: Efficient sparse attention for long-range transformers. arXiv preprint arXiv:2406.16747.

---

> > ### Comment · Reviewer_61qF · 2025-08-04
> >
> > Thank you for taking the time to reply to the review of your work.
> >
> > I am currently unclear on how the statistical testing reported here was performed. I'm slightly concerned by the statement "Correlation with pass@256 performance: Since both RankMe and pass@256 values are nearly monotonically decreasing, the spearman rank correlation is ~1." This suggests these correlational analyses have been run across different model checkpoints rather than across seeds at a single checkpoint. If you are performing statistical testing across different checkpoints for a model I do not believe a spearman correlation is appropriate as this will be reflective of the relationship between training step and performance rather than geometric measure and performance.
> >
> > Thank you for noting that "we will clearly differentiate between established and speculative claims in the figure and note this distinction in the caption." However it remains unclear to me why you would continue to include speculative claims?
> >
> > Thank your for clarification regarding "the relationship between the memorization metric and short-context understanding ability." I still do not believe the experiments are tightly controlled enough to allow clear conclusions about the length of context the model can successfully leverage. As you note "there may also be an effect due to the increased reasoning requirement to answer questions in the TriviaQA dataset." While I appreciate your observing this possibility, it would seem to potentially undermine the core conclusion of this section.
> >
> > Relatedly "we have restricted our analysis to using infini-grams to calculate the memorization metric" - It seems worth clarifying that that infini-grams are in fact quad-grams. You note this in the rebuttal and this seems like an important distinction to be able to understand the paper as written.
> >
> > In the general case, I appreciate the willingness of the authors to engage with the review process. However at this time I still have concerns about the nature of the statistical testing performed - which I believe is key to making the claims this paper hopes to. I also still have concerns about whether the experiments relating to context length are sufficiently well controlled.
> >
> > Overall my review noted that this paper is quite dense, and the rebuttal has left that largely unaddressed. While this work is interesting it currently introduces a lot of ideas for 9 pages, as a result information that seems critical to the reader has been omitted in places (eg. that the infini-gram model is infact a quad gram, or that previous work has suggested SciQ is stable under sliding window attention). I believe this is interesting work but would look forward to a version of it that presents a more focussed, tightly scoped argument better suited to the length of a conference paper --- or a longer format version with sufficient space to properly unpack the broad array of ideas included here.

---

### Official Review · Reviewer_Ts4j · 2025-07-09

**Clarity:** 3
**Significance:** 2
**Originality:** 2
**Rating:** 4
**Confidence:** 3

**Summary:**

This paper explores the evolution of the covariance spectrum of neural network embeddings across training time. During large language model pre-training, three phases are uncovered: a warmup phase wherein the spectral entropy collapses, an entropy-seeking phase where the spectral entropy recovers, and a compression-seeking phase where spectral entropy decreases again. Interestingly, SFT and DPO training are associated with entropy-seeking behavior, whereas RL with verifiable rewards is associated with further compression of the covariance spectrum.

**Questions:**

1. To what extent are there outlier directions in these embedding spaces? Do they affect the analysis at all?
2. Which of your two metrics— RankMe or alpha-ReQ— do you think is more useful and/or appropriate for analyzing embedding geometry? At first glance it seems that alpha-ReQ might be more outlier-resistant, but also makes a distributional assumption about the empirical spectral measure (power law).
3. I think I'm still confused about Figure 5B (bottom). In the caption it sounds like you trained on two different chat datasets, but there is only one chat win rate reported. Is this OOD or ID chat performance?

**Ethical Concerns:**

["NO or VERY MINOR ethics concerns only"]

**Limitations:**

yes

**Quality:**

3

**Strengths And Weaknesses:**

Strengths:
- The discovery that SFT and DPO have the opposite effect on the embedding geometry compared to RLVR is a very interesting result.
- Simple, clear, and timely research question, especially the connection to RLVR.

Weaknesses:
- Did not directly address the possible effect of outlier directions on the analysis. Prior work such as _All Bark and No Bite_ by Timkey and van Schijndel (2021) have found that there are often 1-3 "rogue dimensions" in the residual stream with extremely high variance, which dominate measures like Euclidean distance. Outlier directions, if they exist, would also strongly affect the spectral entropy and RankMe measures, and it is unclear if they should be included or not. Figure 4D does show the top two eigenvalues, but it is unclear what their relationship is to the rest of the spectrum— are they outliers? Direct discussion and investigation of this issue would strengthen the paper significantly (if only to show that outliers are irrelevant here).

---

> ### Author Rebuttal · Authors · 2025-07-31
>
> We thank the reviewer for their insightful comments and suggestions, and we are glad that they found our post-training results interesting and our work to be clear and timely. Below, we address all the concerns and questions raised by the reviewer.
>
> ### Rogue dimensions
>
> The reviewer is correct in noting that rogue dimensions in the LLM representation space affect the RankMe measure, leading to a lower effective rank than the representation dimensionality. This is particularly pronounced in models in the final stages of pretraining, which exhibit a much lower RankMe (computed using the entire eigenspectrum) and therefore indicate an inefficient use of their representation capacity, consistent with Timkey & van Schijndel's findings. In line with the reviewer’s comment, we will add an Appendix figure illustrating the impact of removing these rogue dimensions on RankMe.
>
> In contrast, $\alpha$-ReQ quantifies the decay rate of the eigenspectrum tail, specifically the slope of the power-law decay fit to the eigenspectrum after the top 10 eigenvalues. By disregarding the top (rogue) eigendirections, $\alpha$-ReQ is less influenced by a few rogue directions. However, a strong inverse correlation between the two metrics indicates that the LLM representation space eigenspectrum is significantly skewed beyond just the top rogue dimensions.
>
> Our work demonstrates that analyzing the degree of this skewness using our proposed spectral metrics, and observing its evolution across pretraining and post-training, provides deeper insights into model behavior. We concur with the reviewer that integrating these distinctions between the two metrics and their connection to Timkey & van Schijndel's work into the discussions will enhance our paper, and we propose to implement these changes in the final version.
>
>
> ### $\alpha$-ReQ vs RankMe
>
> While $\alpha$-ReQ is more robust to outlier rogue dimensions than RankMe and is based on an empirically observed power-law distribution of tail eigenvalues, both metrics are strongly anti-correlated and largely interchangeable. However, it is worth noting that RankMe is more sensitive to the ambient dimensionality of the representation space and dataset variations (Fig. 8), making cross-model comparisons challenging (Fig. 9).
>
> ### Fig 4D clarification
>
> We sincerely apologize for this confusion. Figure 4 presents the simplified setting that seeks to verify our analytical results about learning dynamics of cross-entropy loss. In these particular plots (and in Fig 10a), we set the feature dimensionality, $d=2$. Therefore, the eigenspectrum has only 2 eigenvalues, $\sigma_1, \sigma_2$ $\textemdash$ the ones that are plotted in Fig 4D.  We will clarify these points in the updated manuscript.
>
> ### Fig 5B clarification
>
> We apologize for the confusion in these results. The reviewer has correctly noted that we fine-tuned (SFT) models on two different datasets: Anthropic-HH (AH) and AlpacaFarm (AF). Both these sets of models were then evaluated on a third, out-of-distribution (novel) dataset: AlpacaEval.
>
> Moreover, it is worth noting that AlpacaEval is more out-of-domain for AH models than for AF models. Anthropic-HH is a chat dataset focused on helpfulness and harmlessness, with conversational, clarifying, and thought-provoking responses to complex user queries. In contrast, AlpacaFarm and AlpacaEval are generic chat datasets with straightforward, atomic prompts that elicit short, concise responses. The qualitative differences in chat responses and prompt complexity make AlpacaEval a more significant shift for AH models.
>
> The winrate is computed by comparing the AH and AF model responses for the AlpacaEval prompts. This comparison is done with a larger LLM judge which picks the more appropriate answer among the two. As described above, AlpacaEval is more out-of-domain for AH models than for AF models. This is reflected in the low winrates (6-14%) of AH fine-tuned models with respect to AF fine-tuned models. Our key observation is the decrease in this winrate with more pretraining of the base model, suggesting that “overtrained” models (those with higher RankMe values) perform better on in-domain tasks but worse at generalizing to out-of-domain data.
> We will incorporate these clarifications about the experiment design in the final version of our paper.
>
> ---
> We thank the reviewer once again and are grateful for their helpful suggestions. We believe that incorporating discussion of rogue dimensions and the nuances about the two spectral metrics will strengthen our paper and improve its readability. We hope that the reviewer finds our responses to be helpful, and are happy to clarify any further questions during the discussion phase.

---

### Author Response · Authors · 2025-08-09
**Response to All Reviewers (Summary of Changes)**

We sincerely thank all reviewers for their thoughtful and constructive feedback, which has helped us improve the clarity, completeness, and impact of our work. We are encouraged by the positive evaluations and are glad our paper is seen as novel, timely, well-written, relevant, and of interest to the community. We will incorporate clarifications in the main text and add new results in the appendix with appropriate references to them in the main text. Below, we summarize the key changes we propose in response to the reviews.

1. **Effect of Rogue dimensions in spectral metrics (Reviewer Ts4j)**
   We will add eigenspectrum plots and show how RankMe and $\alpha$-ReQ values change with and without including the top eigenvalues for different model families and sizes.

2. **Correlation between spectral metrics and performance (Reviewer 61qF)**
   We will add the tables showing strong correlations between spectral metrics and performance across training (pretraining and RLVR) for the studied model families, and across different initialization seeds for smaller Pythia models. We will also discuss the caveat around studying correlation across seeds (limited variability in spectral metrics and performance) in the main text.

3. **Experimental details for computing spectral metrics (Reviewer 61qF, nzGd)**
   We will specify the number of sequences from the FineWeb dataset and the sequence length used for computing RankMe and $\alpha$-ReQ in the main text, with references to the ablation experiments in the appendix (Fig 7, 8).

4. **Clarification about infini-gram context length (Reviewer 61qF)**
   We will add the table showing the distribution of suffix length used by the infini-gram model for predicting the token probabilities in TriviaQA, and refer to this table in the main text to support our claim that infini-gram likelihoods correspond to information aggregation over short- and medium-context lengths.

5. **Context length dependency of SciQ vs TriviaQA (Reviewer 61qF)**
   We will refer to previous works demonstrating the stability of SciQ performance under sliding window attention and the lack of such behavior for TriviaQA, supporting our claim that the discrepancy in SciQ and TriviaQA performance during the entropy-seeking phase could be related to the requirement of long-context information aggregation ability in TriviaQA but not SciQ.

6. **Control experiment on cross-entropy’s role (Reviewer 61qF)**
   We will add the plots from our analytically-tractable setting when using a MSE-loss showing that cross-entropy loss is essential for the empirically observed non-monotonic phases of representation geometry changes.

7. **Clarification of supervised finetuning setup (Reviewer 61qF, Ts4j)**
   We will update the SFT experimental setup description to clarify the different training setups used and add an intuitive explanation of the results.

8. **Layer-wise spectral metrics of intermediate representations (Reviewer 7SLc, nZGd, YXFK)**
   We will add the plots showing evolution of spectral metrics for intermediate layer representations across different model families, demonstrating how representations change in intermediate layers. We hope that this result will motivate future studies that seek to better understand model internals and the effect of gradient descent in their evolution.

9. **Impact of perturbing specific eigenvectors on downstream tasks (Reviewers Ts4j, YXFK)**
   We will add the table demonstrating the differential importance of the tail of eigenspectrum on downstream tasks, supporting our claim that it is important to use metrics that characterize the whole eigenspectrum rather than only dominant directions.

We appreciate the reviewers’ insights and helpful suggestions. We believe that these clarifications, additional results, and expanded analyses will strengthen the final version of our paper and make it more impactful for the community.

---

### Decision · Program_Chairs · 2025-09-17

**Decision:**

Accept (poster)

**Comment:**

This paper studies the training dynamics geometry of last-token representations in LLMs. The authors use existing metrics that reflect the representation covariance’s geometry, such as the effective rank measure (which is the entropy of the covariance eigenvalues) and power-law decay rate of the eigenvalues. The contribution of this paper is to use the geometric metrics during training to explain the evolution of the last-token representations and to connect it to memorization and generalization. The authors study both the pretraining and the post-training of LLMs.

The authors reveal three geometry-related phases in the pretraining: a warmup phase, an entropy-seeking phase that relates to memorization, and a compression-seeking phase that relates to generalization. In post training, the authors remarkably show that supervised fine-tuning (SFT) and direct preference optimization (DPO) show entropy-seeking dynamics, whereas reinforcement learning with verifiable rewards (RLVR) shows compression-seeking dynamics. While this paper is mainly an empirical analysis of modern LLMs, the authors also provide complementary analytical insights into the training dynamics of a toy model.

The contributions of this work to the research of LLM training dynamics are appreciated. The author rebuttal adequately addressed many of the review comments, as listed in the summary of changes that the authors provided at the end of the rebuttal.

Yet, the AC-Reviewer discussion indicated that some of the author rebuttal to Reviewer 61qF’s concerns was insufficient. Therefore, the following additions that were proposed by the authors should not be added to the final version of the paper: Correlation between spectral metrics and performance (change 2 in the authors’ summary of changes), this includes the experiments from the author rebuttal to Reviewer 61qF that were provided under the section titles of statistical analysis and statistical testing. Besides that, the author rebuttal was adequate, and the authors are requested to implement the rest of the changes listed in their summary of changes.

The set of reviews and all the information throughout the review process (including the rebuttal and discussions) reflect that the contributions of this paper deserve publication and, therefore, my recommendation is to accept this paper.